# Intensified future heat extremes linked with increasing ecosystem water limitation

Jasper M.C. Denissen[1,2,3*], Adriaan J. Teuling[2], Sujan Koirala[1], Markus Reichstein[1], Gianpaolo Balsamo[4,5], Martha M. Vogel[6], Xin Yu[1] and René Orth[1,7].

[1]Department for Biogeochemical Integration, Max Planck Institute for Biogeochemistry, Jena, Germany

[2]Hydrology and Quantitative Water Management Group, Wageningen University, Wageningen, The Netherlands

[3]Research Department, European Centre for Medium-Range Weather Forecasts, Bonn, Germany

[4]Research Department, European Centre for Medium-Range Weather Forecasts, Reading, United Kingdom

[5]World Meteorological Organization, Geneva, Switzerland

[6]Red Cross Red Crescent Climate Centre, The Hague, The Netherlands

[7]Faculty of Environment and Natural Resources, University of Freiburg, Freiburg, Germany

*Correspondence to*: Jasper M.C. Denissen (jasper.denissen@bgc-jena.mpg.de)

**Abstract.** Heat extremes have severe implications for human health, ecosystems and the initiation of wildfires. Whereas they are mostly introduced by atmospheric circulation patterns, the intensity of heat extremes is modulated by terrestrial evaporation associated with soil moisture availability. Thereby, ecosystems provide evaporative cooling through plant transpiration and soil evaporation, which can be reduced under water stress. While it has been shown that regional ecosystem water limitation is projected to increase in the future, the respective repercussions on heat extremes remain unclear.

In this study we use projections from twelve Earth system models to show that projected changes in heat extremes are amplified by increasing ecosystem water limitation in regions across the globe. We represent ecosystem water limitation with the Ecosystem Limitation Index (ELI) and quantify temperature extremes through the differences between warm-season mean and maximum temperatures. We identify hotspot regions in tropical South America and across Canada and Northern Eurasia where relatively strong trends towards increased ecosystem water limitation jointly occur with amplifying heat extremes. This correlation is governed by the magnitude of the ELI trends and the present-day ELI which denotes the land-atmosphere coupling strength determining the temperature sensitivity to evaporative cooling. Many regions where ecosystem functioning is predominantly energy-limited or transitional in present climate exhibit strong trends towards increasing water limitation and simultaneously experience the largest increases in heat extremes. Sensitivity of temperature excess trends to ELI trends is highest in water-limited regions, such that in these regions relatively small ELI trends can amount to drastic temperature excess

trends. Therefore, considering the ecosystem's water limitation is key for assessing the intensity of future heat extremes and their corresponding impacts.

**Short summary**

Heat extremes have severe implications for human health and ecosystems. Heat extremes are mostly introduced by large-scale atmospheric circulation but can be modulated by vegetation: Vegetation with access to water uses solar energy to evaporate water into the atmosphere. Under dry conditions, water may not be available, suppressing evaporation and heating the atmosphere. Using climate projections, we show that regionally less water is available for vegetation, intensifying future heat extremes.

# 1 Introduction

Heat extremes affect ecosystems and society through their implications on human health, crop yields and tree mortality, and the initiation of wildfires (Anderegg et al., 2013; Goulart et al., 2021; McDowell & Allen, 2015; O et al., 2020; Orth et al., 2022; Ruffault et al., 2020; Vogel et al., 2019). In the recent past, temperature extremes have increased in intensity, duration and frequency; these changes are related to climate change (Seneviratne et al., 2021) and they have even accelerated in recent years in many regions (Seneviratne et al., 2014). In the future, heat extremes are projected to intensify further, alongside the ongoing global warming (Seneviratne et al., 2021).

Hot temperatures can be fueled by dynamic and thermodynamic processes (Harrington et al., 2019; Trenberth et al., 2015). The relevance of atmospheric dynamics for recent heat waves has been highlighted for the case of large-scale blocking patterns which support heat accumulation across consecutive dry days (Cassou et al., 2005; Jézéquel et al., 2018) as well as the entrainment of warm air aloft (Miralles et al., 2014). Also, large-scale circulation patterns advecting warm air, or air from regions with dry soils, have been suggested to contribute to heat waves (Schumacher et al., 2019). Additionally, thermodynamic processes can amplify heat extremes; the land surface determines the partitioning of incoming radiative energy into sensible heating and latent heat (Seneviratne et al., 2010). Changes in this flux partitioning can be induced through soil moisture drying as water-stressed vegetation tends to reduce transpiration; this way, a larger fraction of the incoming energy is available for sensible heating which can lead to elevated temperatures (Budyko, 1974; Denissen et al., 2021; Vogel et al., 2017). As a consequence, circulation-induced rainfall deficits are translated by ecosystem water limitation to reduced evaporative cooling and amplified local temperatures (Miralles et al., 2012; Quesada et al., 2012; Teuling et al., 2010; Ukkola et al., 2018).

It has been shown that climate change may involve regional long-term trends in soil moisture and land-atmosphere coupling (Berg et al., 2017; Berg & Sheffield, 2018; Denissen et al., 2022; Seneviratne et al., 2021; Sippel et al., 2017) and that these

can contribute to amplified heat extremes (Lorenz et al., 2016; Seneviratne et al., 2006; Vogel et al., 2017) especially in the case of depletion of soil moisture preceding the warm season (Rasmijn et al., 2018; Stegehuis et al., 2021). In this study, we revisit and complement this previous research with novel indices and by analyzing output from the latest generation of Earth System models from the Coupled Model Intercomparison Project Phase 6 (CMIP6) (Eyring et al., 2016). In particular we use

(i) a recently introduced ecosystem water stress index: the Ecosystem Limitation Index, or ELI (Denissen et al., 2020). This is a correlative index that evaluates directly the importance of water versus energy stress for terrestrial evaporation, thereby moving beyond the nonlinear relationship between soil moisture and evaporative cooling alone. Further, as this index directly captures evaporative cooling, it links more mechanistically with heat waves than general aridity or land-atmosphere coupling indices. Thereby other factors affecting water-limitation can be functionally addressed (e.g. groundwater, hydraulic failure as

lag effect, $CO_2$). Further, the ELI can be used to pinpoint regime transitions, as positive values are indicative of water-limited conditions, while negative values denote ecosystem energy limitation. In addition, for analyzing heat extremes, we (ii) focus on the difference between warm-season mean and maximum temperatures, hereafter referred to as temperature excess. While temperature excess is known to be affected by land-atmosphere coupling (Dirmeyer et al., 2021; Donat et al., 2017; Lorenz et al., 2016; Schwingshackl et al., 2018; Seneviratne et al., 2006; Sippel et al., 2017; Ukkola et al., 2018; Vogel et al., 2017), the

average temperature is largely driven by large-scale circulation (Cassou et al., 2005; Miralles et al., 2014; Schumacher et al., 2019). This way, we assume that by focusing on the difference between mean and maximum temperatures, we can isolate the thermodynamic component from the dynamic component in heat wave development. As such, we jointly assess trends in ecosystem water limitation and heat extremes in fully coupled CMIP6 simulations from twelve state-of-the-art Earth system models at the monthly time scale and 2˚x2˚ spatial resolution from 1980 – 2100 (Eyring et al., 2016) in order to determine the

thermodynamic contribution of the land surface for present and future heat extremes.

## 2 Materials and Methods

### 2.1 Ecosystem Limitation Index

The Ecosystem Limitation Index (ELI), formerly referred to as the correlation-difference metric (Denissen et al., 2020), is adapted as follows:


Eq. 1) ELI = cor(SM',ET') - cor($T_a$' | $SW_{in}$',ET')

The prime denotes monthly anomalies of root-zone soil moisture (SM), terrestrial evaporation (ET), air temperature ($T_a$) and incoming shortwave radiation ($SW_{in}$). cor(SM',ET') is a proxy for water limitation, whereas cor($T_a$' | $SW_{in}$',ET') is a proxy

for energy limitation. In this context, the | indicates the use of either $T_a$ or $SW_{in}$ anomalies in the second term on the right hand side of Eq. 1, as ET in some regions is limited more strongly by lack of incoming shortwave radiation (Nemani et al., 2003) and in other regions more strongly by cold temperatures. Therefore, we test for each grid cell which energy proxy yields the

highest correlation with ET (cor($T_a$',ET') vs. cor($SW_{in}$',ET')), and is hence most relevant in this location, to then use it in the computation of ELI in the respective grid cell (Supplementary Figure 1). Between energy- and water-limited conditions, the

ELI expresses different typical sensitivities to energy and water supply: High and positive cor($T_a$' | $SW_{in}$',ET') is indicative of energy-limited conditions, whereas high and positive cor(SM',ET') indicates water-limited conditions. The ELI combines both the relevance of energy and water supply for evaporative cooling by taking the difference between those two correlations, so that positive values denote water-limited conditions and negative values indicate energy-limited conditions. Thereby, the ELI can be used to pin-point transitional areas where regime shifts occur frequently, where ELI is approximately zero. Further, in

contrast to other traditional indices, such as the Aridity Index, that rely on climatological means, the ELI can be used to study (parts of) the seasonal cycle. For a more extensive assessment of air temperature or incoming shortwave radiation and soil moisture as the choices for energy and water proxies as well as a detailed elaboration on the interpretation of ELI, please refer to Denissen et al. (2022).

**2.2 CMIP6 data**

In this study, we use data from the Coupled Model Intercomparison Project (CMIP6) (Eyring et al., 2016), of which the most important information on the used data is summarized in Table 1. We only selected models that provide i) historical (1980 - 2015) and "worst-case" SSP5-8.5 (2015 – 2100) (O'Neill et al., 2016) simulations, ii) the necessary variables (Table 1) and iii) sufficient spatial (2˚x2˚ or finer grid cell resolution) and temporal (monthly) resolutions. The maximum daily temperature

denotes the maximum daily average temperature per month. By taking the SSP5-8.5 scenario we intend to focus on the climate scenario most influenced by human activity and related emissions of greenhouse gasses.

Table 1. Overview of model details and model output used in this study. The following variables have been downloaded from all the models at the monthly time scale: temperature (tas), the total water content per soil layer (mrsol), terrestrial evaporation

(hfls), leaf area index (lai), maximum daily temperature (tasmax) and in- and outgoing short- and longwave radiation (rsds,rsus,rlds,rlus). Dynamic vegetation reflects whether or not plant functional traits (PFT) can vary in time, responding to competition for resources. These resources could but do not necessarily include any combination of nitrogen, phosphorus, water and energy. However, the resources considered in this context vary between models. As land use change forcing is identical for all models for the SSP5-8.5 scenario (O'Neill et al., 2016), this column only concerns historical simulations. For

historical simulations, land use change forcing comes from the Land Use Harmonization (LUH) 2 v2h product (https://luh.umd.edu/data.shtml) (Hurtt et al., 2011), except if mentioned otherwise. As land cover types might vary between models, land use change forcing effects might differ as well. *: in the CMIP6 members, or variants, differences exist in the forcing index (f). This index number indicates the forcing used for the respective realization and can be used to distinguish

between CMIP6-recommended or other forcing data sets. Which forcing dataset f represents is defined per model. \*\*: the first

number denotes the version of the historical simulation, whereas the second number indicates the SSP5-8.5 simulation.

| Institution | Model | Member* | Version** | Dynamic vegetation | Irrigation | Land use change | Citation |
|---|---|---|---|---|---|---|---|
| Commonwealth Scientific and Industrial Research Organisation (CSIRO) | ACCESS-ESM1-5 | r1i1p1f1 | v20191115 & v20191115 | yes | no | yes | (Ziehn et al., 2019a, 2019b, 2020) |
| Beijing Climate Center (BCC) | BCC-CSM2-MR | r1i1p1f1 | v20181126 & v20190314 | no | no | yes, explicitly involved in BCC-AVIM2.0 | (Wu et al., 2018, 2019; Xin et al., 2019) |
| Centro Euro-Mediterraneo sui Cambiamenti Climatici (CMCC) | CMCC-ESM2 | r1i1p1f1 | v20200622 & v20200622 | yes | no | yes | (Cherchi et al., 2019; Lovato & Peano, 2020a, 2020b) |
| Centre National de Recherches Météorologiques (CNRM) | CNRM-CM6-1 | r1i1p1f2 | v20190410 & v20190410 | no | no | yes | (Voldoire, 2018, 2019a; Voldoire et al., 2019) |
| CNRM | CNRM-ESM2-1 | r1i1p1f2 | v20181206 & v20191021 | no | no | yes | (Seferian, 2018; Séférian et al., 2019; Voldoire, 2019b) |

| EC-Earth-Consortium | EC-Earth3-CC | r1i1p1f1 | v20210113 & v20210113 | yes | Indirectly, through irrigated crop | yes | (Consortium (EC-Earth), 2021a, 2021b; Döscher et al., 2021) |
|---|---|---|---|---|---|---|---|
| National oceanic and Atmospheric Administration (NOAA), Geophysical Fluid Dynamics Laboratory (GFDL) | GFDL-ESM4 | r1i1p1f1 | v20190726 & v20180701 | yes | no | yes | (Dunne et al., 2020; John et al., 2018; Krasting et al., 2018) |
| Met Office Hadley Centre (MOHC) | HadGEM3-GC31-LL | r1i1p1f3 | v20200114 & v20190624 | yes | no | yes | (Good, 2020; Ridley et al., 2019; Williams et al., 2018) |
| Max Planck Institute for Meteorology (MPI-M) | MPI-ESM1-2-HR | r1i1p1f1 | v20190710 & v20190710 | no | no | yes | (Jungclaus et al., 2019; Mauritsen et al., 2019; Müller et al., 2018; Schupfner et al., 2019) |

| MPI-M | MPI-ESM1-2-LR | r1i1p1f1 | v20190710 & v20190710 | yes | no | yes | (Mauritsen et al., 2019; Wieners, et al., 2019; Wieners, et al., 2019) |
|---|---|---|---|---|---|---|---|
| Meteorological Research Institute (MRI) | MRI-ESM2-0 | r1i1p1f1 | v20190222 & v20191108 | no | no | yes | (Yukimoto, Kawai, et al., 2019; Yukimoto, Koshiro, et al., 2019a, 2019b) |
| MOHC | UKESM1-0-LL | r1i1p1f2 | v20190627 & v20190726 | yes | no | yes, for crops and pasture. | (Good et al., 2019; Sellar et al., 2019; Tang et al., 2019) |

## 2.3 Pre-processing data

All data is regridded to a common 2˚x2˚ grid cell resolution using bilinear interpolation after applying a model-specific land-sea mask. After data acquisition, several steps are taken to assure a meaningful selection of data for the analysis. First, to pin-point the hottest heat extremes, we focus on the three hottest months a year (warm season), defined as the 3 months-of-year with the highest maximum daily temperature averaged decadally. The advantage of considering only the warm season lies in the comparison of concomitant trends of ELI, evaporative fraction (EF) and temperature excess, as these might be subject to seasonal variability. Second, to additionally assure that we are investigating the active vegetation periods during the warm season, which would elicit vegetation responses to anomalies in energy and water supply affecting the surface flux partitioning, all months with $T_a < 10$˚C and Leaf Area Index (LAI) $< 0.2$ $m^2$ $m^{-2}$ are excluded from the analysis. Thereby, we disregard mainly grid cells in the most sparsely vegetated regions in Northern Africa and Western China and cold regions in the Northern latitudes, but retain major drylands including parts of the Sahel and the Australian interior (Supplementary Figure 2). This selection of data results in what we refer to in this manuscript as the "warm vegetated land area". Further, root-zone soil moisture is computed as a weighted average of the total water content per soil layer present in the top meter of soil. This data

is then used to compute the decadal time series of the desired diagnostics, which are ELI, EF and temperature excess. EF is computed as the fraction of the net surface radiation (the sum of all radiative components) that is used to evaporate water. Temperature excess is computed for each grid cell and decade as the difference between the means of (i) the 10 warm-season mean temperatures from the individual years and (ii) the 10 temperature maxima in the individual years. Next to this, we assess ecosystem water limitation with the ELI (Equation 1) (Denissen et al., 2020).


## 2.4 ERA5-Land analysis

Reanalysis data, including the variables 2m temperature, soil moisture layers 1-3, latent heat flux, LAI for high and low vegetation and downward solar radiation, from ERA5-Land from 1950 – 2020 were used to validate the CMIP6-based results (Muñoz Sabater, 2019; Muñoz-Sabater et al., 2021). All data has been aggregated to the monthly time scale and 2˚x2˚ spatial

resolution. Maximum daily temperature was computed as the maximum average daily temperature per month. The root-zone soil moisture encompasses the soil moisture in top meter of the soil and is computed as a weighted average of soil moisture layers 1 (0 – 7cm), 2 (7 – 28cm) and 3 (28 – 100cm). The same methodology as has been applied to the CMIP6 data to compute temperature excess and ELI has been applied to the reanalysis data. Vegetated conditions were assumed when the LAI of either high or low vegetation > 0.2.


## 2.5 Computing Theil-Sen slopes and slope significance

The trends shown in Figure 1, 2 and 6 and Supplementary figures 3, 4 and 5 are based on Theil-Sen slopes (Sen, 1968; Theil, 1992). This approach is insensitive to statistical outliers, as the median slope from a range of slopes through all pairs of points is selected as the best fit. The significance of these slopes is determined based on Kendall's tau statistic from Mann-Kendall

tests.

## 3 Results

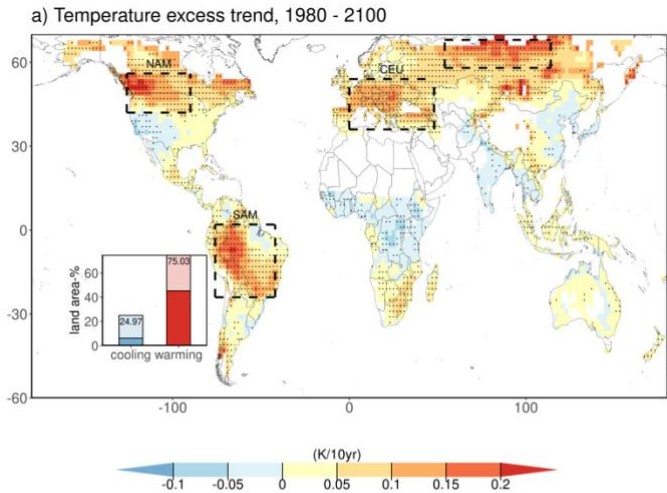

a) Temperature excess trend, 1980 - 2100

(K/10yr)

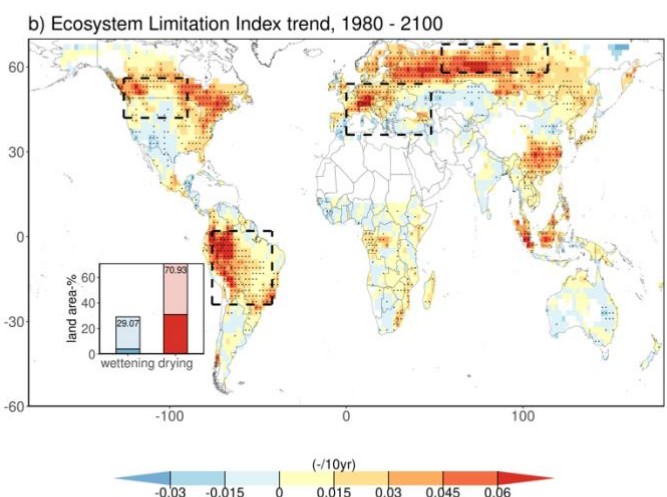

b) Ecosystem Limitation Index trend, 1980 - 2100

(-/10yr)

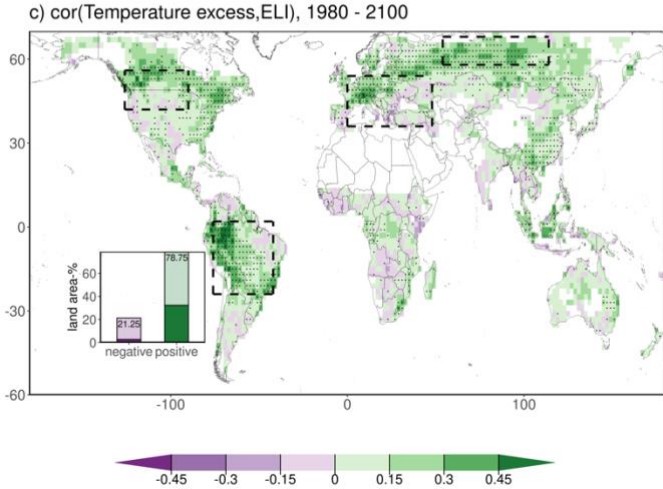

c) cor(Temperature excess,ELI), 1980 - 2100

**Figure 1.** Similarity of global patterns of change in temperature excess and ecosystem water limitation. Multi-model means of trends based on decadal time series per respective CMIP6 model of a) temperature excess) and b) Ecosystem Limitation Index (ELI). c) Multi-model means of Kendall's rank correlation coefficient between model-specific time series of ELI and temperature excess. The insets display the fraction of the warm land area with positive or negative trends or correlations, respectively (at least 8 out of 12 models agreeing on the sign of the trend or correlation are hued darker). Stippling indicates that at least 8 out of 12 CMIP6 models agree on the sign of the trend or correlation. All trends and correlations are calculated over the warm season and are only displayed if at least 8 CMIP6 models have full time series available, such that white areas denote regions with no or insufficient data. The dashed boxes indicate regions of interest, which are regions where temperature excess increases are particularly rapid and spatially coherent: North and South America (NAM and SAM), Central Europe (CEU) and Northern Asia (NAS).

We identify increased temperature excess trends across over 75% of the warm vegetated land area from 1980 - 2100 (Figure 1a). Model confidence is higher for increasing than for decreasing temperature excess (inset plot Figure 1a), as in almost half of the area with increasing temperature excess at least eight out of twelve CMIP6 models agree, while this is much less for decreasing temperature excess (see also Supplementary Figure 3). This reveals high confidence in an accelerated increase of heat extremes compared with warm-season mean temperatures.

There is a widespread increase in incoming shortwave radiation in about 71% of the warm vegetated land area, with high inter-model agreement (Supplementary Figure 4), which can directly affect near-surface temperature through the surface energy balance. These trends could result from projected decreases in aerosol emissions (Nabat et al., 2014), or from changes in cloud cover. As daily maxima of incoming shortwave radiation roughly co-occur with daily temperature maxima, increased incoming shortwave radiation links more strongly to increased in maximum temperatures rather than mean temperatures (Qian et al., 2011), which are more strongly governed by the longwave radiation budget.

ELI increases in more than 71% of the warm vegetated land area (Figure 1b), signaling shifts towards water limitation. Generally, models particularly agree on the sign of the ELI increases (stippling in Figure 1b), whereas more uncertainty exists with respect to the magnitude of ELI trends (Supplementary Figure 5). Further, we note that in the mid- to high latitudes, ELI trends are generally temperature controlled, whereas the tropics are more sensitive to incoming shortwave radiation (Supplementary Figure 1), thereby acknowledging and allowing that energy proxies can vary locally.

Spatial patterns of multi-model mean trends in temperature excess and ELI are very similar. Areas with the highest temperature excess trends (>0.2 K/10yr) are exclusively characterized by ELI increases. More importantly, also the temporal evolution of decadal time series of temperature excess and ELI is similar in many regions. This is evidenced by significant correlations in many areas (Figure 1c, Supplementary Figure 6), suggesting that increasing ELI contributes to hotter temperature extremes.

As correlations cannot distinguish the direction of causality, we stress that hotter temperature extremes can in turn further dry out terrestrial vegetation, thereby increasing water limitation. Additionally, heat extremes and related hydraulic failure could lead to plant mortality (McDowell & Allen, 2015), limiting evaporative cooling even more. As such, these pathways further strengthen positive correlations between ELI and temperature excess. We also find regions with insignificant and even negative correlations such as parts of the Sahel, Kazakhstan, the Balkan, North America and Southern Africa. As plant transpiration scales with LAI, this limits the ability of the scarce vegetation present in such regions to provide sufficient evaporative cooling, possibly rendering correlations insignificant. Further deviations from a positive relationship between temperature excess and ELI might result from alternative processes such as (changes in) advection of warm air masses through large-scale circulation patterns, while positive relationships could be exaggerated by changes in incoming shortwave radiation (Supplementary Figure 4).

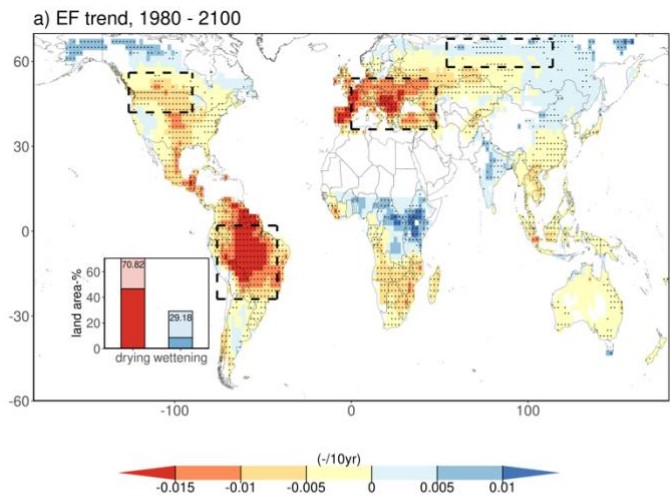

a) EF trend, 1980 - 2100

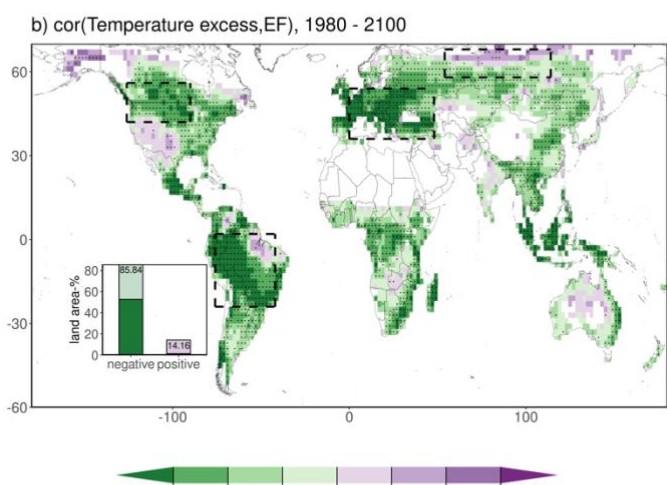

b) cor(Temperature excess,EF), 1980 - 2100

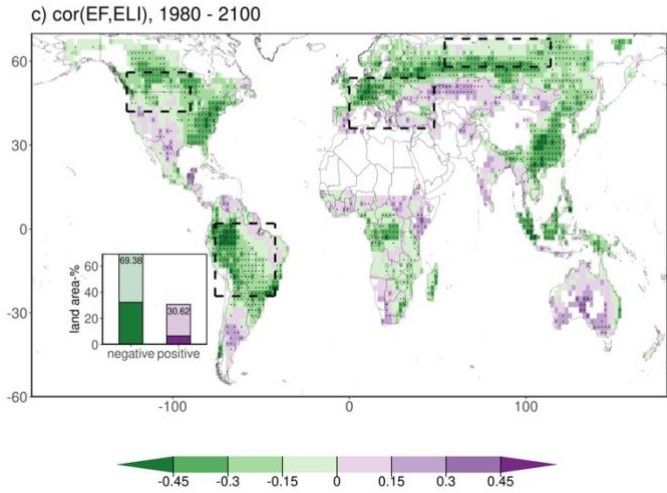

c) cor(EF,ELI), 1980 - 2100

**Figure 2.** Global multi-model mean distribution and trends of Evaporative Fraction (EF). Multi-model mean of trends based on decadal time series per respective CMIP6 model of a) EF and b) Ecosystem Limitation Index (ELI). c) Multi-model mean of Kendall's rank correlation coefficient between model-specific time series of ELI and temperature excess. The insets display the fraction of the warm land area that with positive or negative trends or correlations, respectively (at least 8 out of 12 models agreeing on the sign of the trend or correlation are hued darker). Stippling indicates that at least 8 out of 12 CMIP6 models agree on the sign of the trend or correlation. All trends and correlations are calculated over the three hottest months-of-year, defined as the 3 months–of-year which have the highest average temperature over 1980 - 2100. The dashed boxes indicate regions of interest.

Furthermore, in order to illustrate the physical link between ELI and temperature excess, which presumably is through evaporative cooling, we analyze terrestrial evaporation normalized by net surface radiation. The resulting EF links the surface energy and water balances. The EF is decreasing in all regions of interest but Northern Eurasia, with high agreement between individual models (Figure 2a). Moreover, EF is generally significantly correlated with both temperature excess and ELI, respectively, suggesting the physical link between these quantities. This way, in approximately 86% of the warm vegetated land area, trends in EF fraction are negatively correlated with temperature excess, meaning that a decreasing (increasing) trend in EF, renders more (less) energy available for sensible heating, which elevates (reduces) heat extremes (Figure 2b). In about 69% of the warm vegetated land area, the correlation between EF and ELI is negative (Figure 2c), verifying that most shifts towards ecosystem water limitation jointly occur with the expected decreases in evaporative cooling. Some regions, such as central US, the Mediterranean and Northern Mongolia, exhibit insignificant or even positive correlations, possibly pointing to other processes such as irrigation and/or land use changes (Table 1).

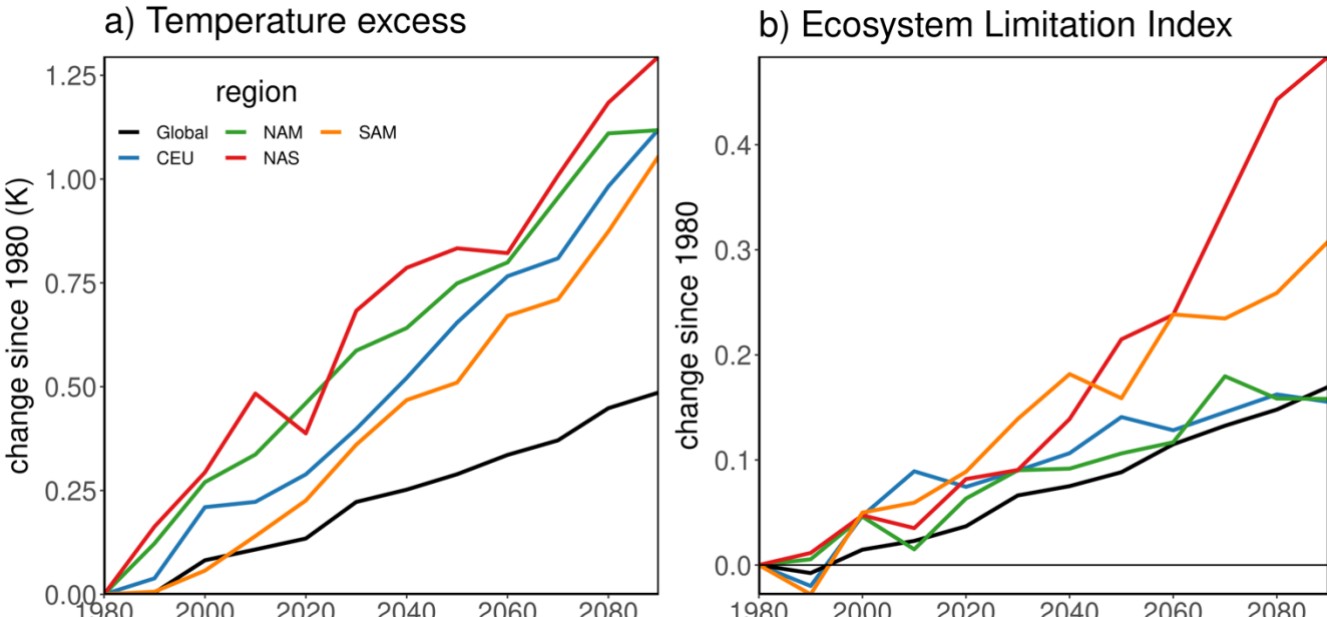

**Figure 3.** Changes in global and regional temperature excess with increasing ecosystem water limitation. Temporal evolution of a) temperature excess and of b) Ecosystem Limitation Index (ELI) globally and for the regions of interest. Solid lines depict multi-model mean time series. Global and regional averages are calculated over land grid cells that have complete time series for all models and variables and are weighted according to the surface area per grid cell.

Next, we compare the temporal evolution of temperature excess and ELI averaged across the regions of interest and the entire warm vegetated land area between historical and future time periods. Figure 3a shows a steady global increase of temperature excess, with warm-season maximum temperature experiencing an additional 0.5K warming with respect to the average warm-season temperature over $1980 - 2100$. In all regions of interest, temperature excess is increasing over twice as fast as the global average. Even though uncertainty in temperature excess exists between individual models (Supplementary Figure 3 and 7a), the majority of models agree both globally and regionally that temperature excess is significantly increasing.

ELI trends differ more strongly in magnitude across the regions of interest than the temperature excess trends (Figure 3b). While underlying ELI trends from individual models generally tend to display positive ELI trends, there is a larger spread both in magnitude and in sign (Supplementary Figure 7b). This indicates different contributions of the ELI to the temperature excess trends between models (Supplementary Figure 6) and regions; while the ELI contribution is particularly strong in NAS and SAM, as can also be seen from the correlations in Figure 1c, it is weaker but still considerable in CEU and NAM where probably other processes play a role such as changes in large-scale circulation patterns or boundary layer dynamics. Further,

most significant trends in Supplementary Figure 7b are positive, underlining a higher confidence of the model ensemble to
250  project increasing rather than decreasing ecosystem water limitation.

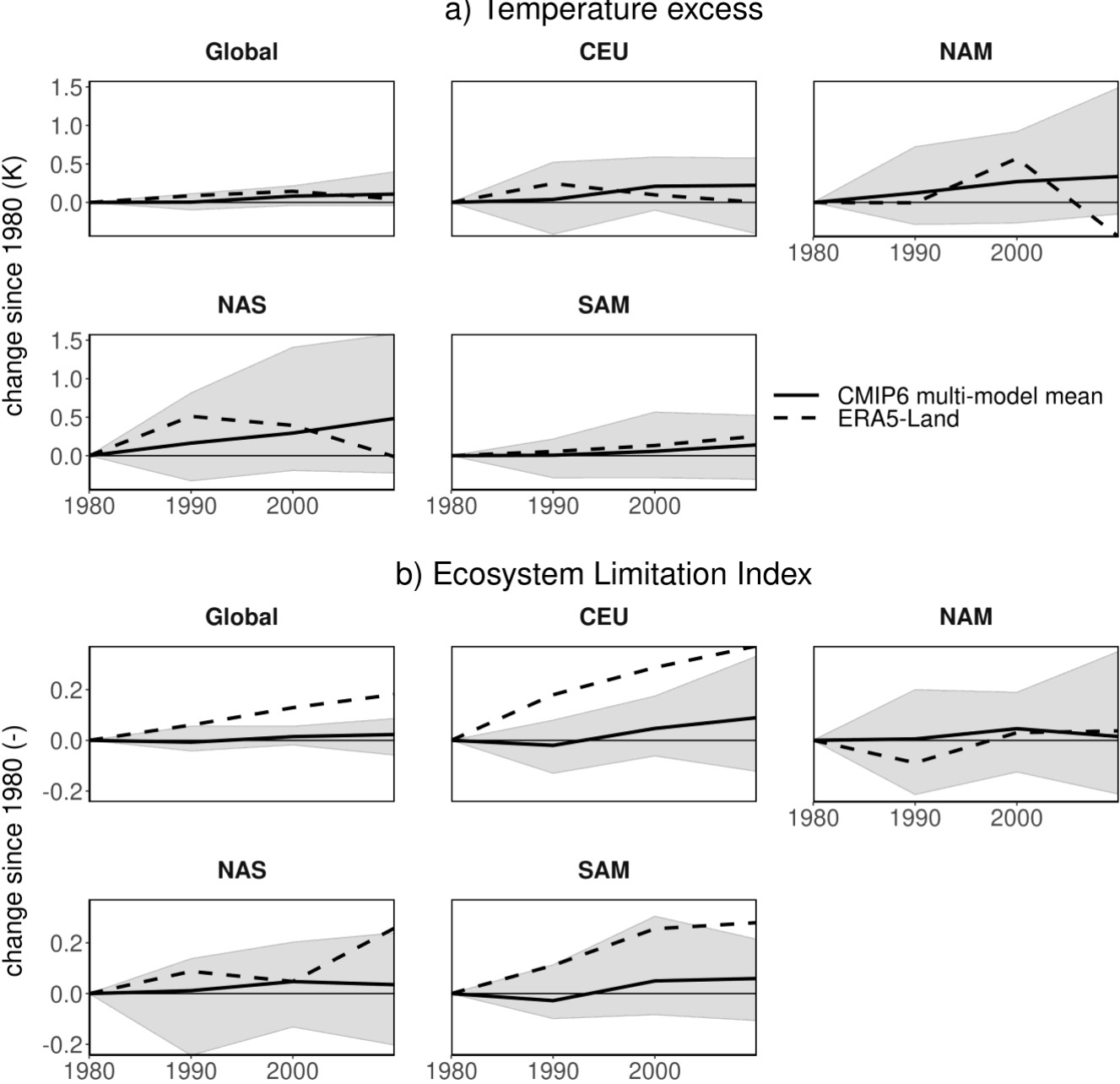

**Figure 4.** Changes in global and regional temperature excess in concert with increasing ecosystem water limitation from CMIP6 models and ERA5-Land. Temporal evolution of a) temperature excess and of b) Ecosystem Limitation Index (ELI) globally and for the regions of interest. The black solid lines depict global and regional time series from the CMIP6 models, while the black dashed line represents ERA5-Land. The grey ribbon displays the envelope which encapsulates all the CMIP6 results. Global averages are calculated over land grid cells that have complete time series for all models and variables and are weighted according to the surface area per grid cell. The same mask is applied for CMIP6 models and ERA5-Land.

During 1980 – 2020, temperature excess computed from ERA5-Land data lies largely within the envelope of the individual CMIP6 models (Figure 4a). As such, the temperature excess findings from individual CMIP6 models are not implausible. As the ERA5-Land dataset is supported by the comprehensive assimilation of available observations, the similarity of the CMIP6 model results in terms of temperature excess demonstrates a successful validation of the models considered here. This is further corroborated by surface air temperature extremes from CMIP5 and CMIP6, that compare well with observation-based data sets, albeit with model-specific performance that varies in space and time (Thorarinsdottir et al., 2020). At the same time, the CMIP6-based ELI is only partly corroborated by the ERA5-Land reanalysis data from 1980 – 2020 (Figure 4b), as globally and in half the regions of interest the reanalysis-based ELI exceeds the CMIP6 envelope. In this historical time period and across most regions of interest, the CMIP6 trends for both temperature excess and ELI are generally more positive than negative, which corroborates a positive relationship between the two, as is also seen further into the future (Figure 3). This relationship is weaker in the observation-based estimate from ERA5-Land, where temperature excess mostly stays within the multi-model envelope and only increases monotonically in SAM, while ELI exceeds the multi-model envelope and increases in all regions of interest except NAM. This indicates a different coupling between ELI and temperature excess in ERA5-Land than in the CMIP6 models, which should be further investigated in the future. Note that ERA5-Land is only indirectly supported by data assimilation, as meteorological forcing from ERA5 assimilates observations only for 2m temperature, relative humidity and surface soil moisture. Therefore, temperature excess benefits more directly from data assimilation than ELI, which is based on ET and (root-zone) soil moisture which are not readily observed across the globe. This way, ERA5-Land estimates of the global ELI evolution are subject to uncertainty, and while it provides an independent reference for comparing the CMIP6 model results it is itself based on the land surface model dynamics underlying the ERA5-Land dataset. Next to that, differences could arise due to different land cover maps underlying respective simulations from ERA5-Land and the CMIP6 models.

The tendency of temperature excess to be elevated in response to increasing ecosystem water limitation becomes even clearer when only grid cells where at least eight out of twelve CMIP6 models agree on the sign of the temperature excess trends are included. This is evidenced by a stronger increase of ELI in regions with robust temperature excess trends (Supplementary Figure 8). ELI trends are even larger for regions with robust and positive temperature excess trends. At the same time no clear trends in ELI are found for regions with robust and negative temperature excess trends. This suggests that factors other than evaporative cooling, such as changes in circulation, render the temperature excess trends negative in these regions.

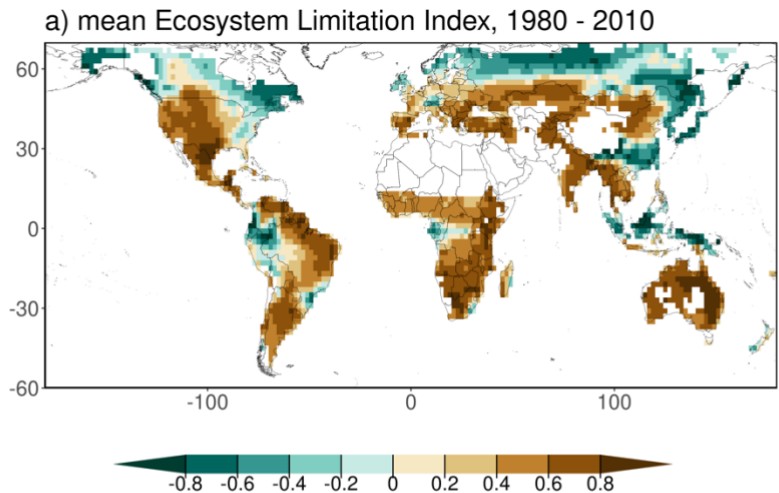

a) mean Ecosystem Limitation Index, 1980 - 2010

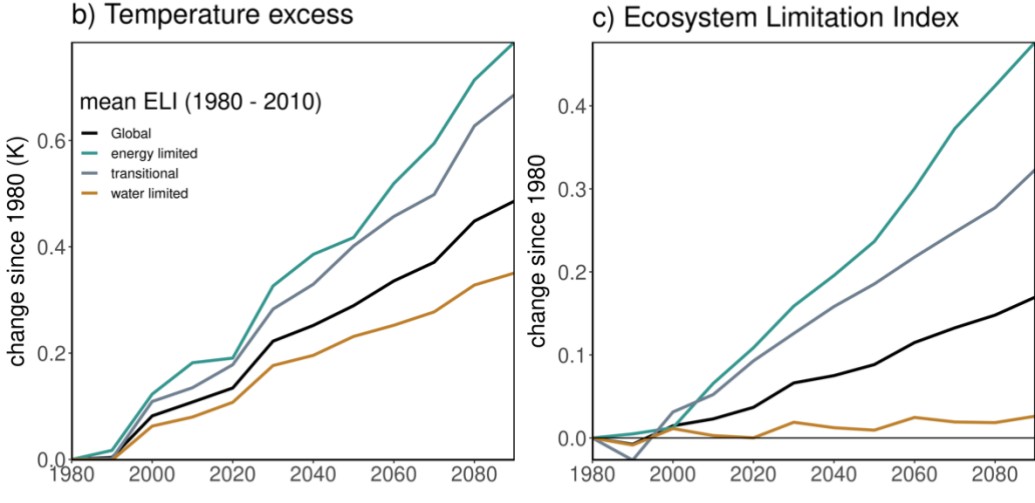

b) Temperature excess

mean ELI (1980 - 2010)
— Global
— energy limited
— transitional
— water limited

c) Ecosystem Limitation Index

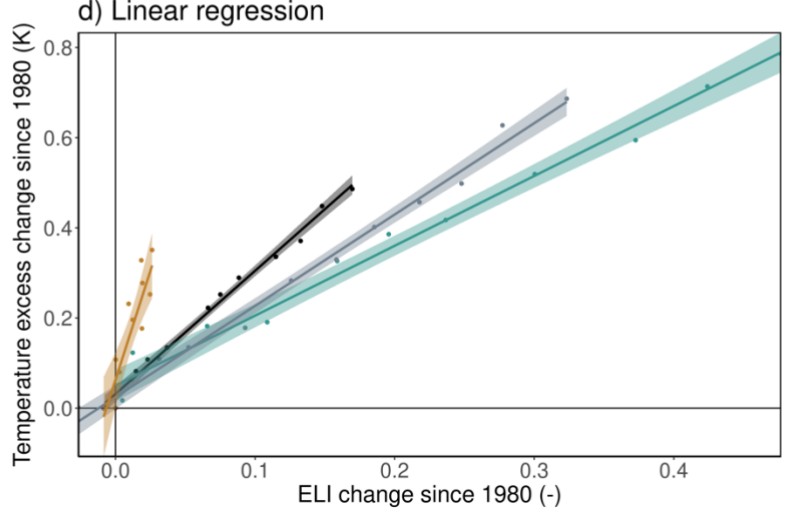

d) Linear regression

**Figure 5.** Relation between temperature excess and ecosystem water limitation. a) Multi-model mean Ecosystem Limitation Index (1980 - 2010). Solid lines depict the time series of multi-model means inferred from globally (black) and regionally (colored) decadally averaged model simulations for b) temperature excess and c) Ecosystem Limitation Index. The classification is defined based on the model-specific mean ELI over 1980 - 2010 (Supplementary Figure 9): Energy limited (ELI < -0.2), transitional (-0.2 < ELI < 0.2) and water limited (ELI > 0.2). d) Points denote the global (black) and regional (colored) decadal multi-model means of ELI (x-axis) and temperature excess (y-axis), expressed as change since 1980. The lines denote linear regressions, with a shaded colored 95% confidence interval. Land grid cells that do not have complete time series for all models are excluded (white regions, Methods). Global and regional averages are weighted according to the surface area per grid cell.

The sensitivity of temperature excess to ELI trends is expected to depend on the initial regime and can be explained through the nonlinear relationship between soil moisture and EF (Denissen et al., 2022; Seneviratne et al., 2010): In initially energy-limited grid cells (soil moisture exceeds critical soil moisture), ecosystems can sustain maximum EF, assuming sufficient available energy during the warm season. Hence, in such grid cells shifts towards water limitation, expressed by positive ELI trends or soil drying, do not amount to large changes in surface flux partitioning, nor in temperature excess, resulting in low sensitivity between ELI and temperature excess trends. In initially water-limited grid cells (soil moisture below critical soil moisture), further soil drying, or shifts towards water limitation, can reduce EF. This way, temperature excess trends are highly sensitive to ELI trends in water-limited grid cells. Transitional grid cells, which are characterized by a soil moisture regime that transitions periodically from below to above the critical moisture content, effectively switch between energy- and water-limited conditions frequently. As such, evaporative cooling and consequently temperature excess are periodically sensitive to increasing water limitation. In extremely dry and water-limited conditions, where soil moisture values approach the wilting point, hardly any moisture can be extracted from the soil, rendering vegetation activity and associated EF too low to provide ample evaporative cooling. As such, shifts towards ecosystem water limitation should hardly decrease evaporative cooling further in extremely water-limited grid cells. To test this hypothesis, we classify all grid cells based on their respective mean ELI over 1980 - 2010 (Figure 5a) to define energy-limited (ELI < -0.2), transitional (-0.2 < ELI < 0.2) and water-limited (ELI > 0.2) conditions. We analyze temperature excess trends across these three regimes and find that over initially water-limited areas they are below the global average, while trends over initially transitional or energy-limited areas are above the global average (Figure 5b). This is against our initial expectation but can be explained by the corresponding ELI trends which are much more pronounced in energy-limited regions (Figure 5c), leading to more often occurring water-limited conditions in these areas. In initially water-limited regions, temperature excess increases despite only marginal ELI increases over the study period, possibly pointing a higher sensitivity of temperature excess to ELI increases in such regions. Moving beyond trends we also analyze the sensitivity of decadal temperature excess with respect to ELI for energy-limited vs. transitional vs. water-limited areas and find the strongest relationship in the case of water-limited areas (Figure 5d), as evidenced by the largest increase in temperature excess with ELI. This confirms that changes in water-limited areas temperature excess trends are most

sensitive to ELI trends. This stresses that evaporative cooling in already arid drylands is even further reduced, increasingly limiting their ability to mitigate future heat extremes (Feldman et al., 2023). Despite lower sensitivity in transitional and energy-limited regions, ELI trends and related reductions in evaporative cooling are much larger, amounting to larger temperature excess trends.

To quantify the strength of the relationships displayed in Figure 5d we compute correlations for the relationships shown for the three regimes, respectively (crosses in Supplementary Figure 10a). This suggests a more robust link between ELI and temperature excess in transitional and energy-limited areas resulting from the strong ELI trends moving these areas towards water-limitation. To study the relevance of spatial variability across the grid cells that are initially energy- or water-limited or transitional for the correlation estimates, the grid-specific time series of temperature excess and ELI are bootstrapped and displayed as boxplots in Supplementary Figure 10a, with overall similar results. Whereas sensitivity in water-limited regions in Figure 4d is higher, more uncertainty exists in its relationship, as evidenced a larger spread of bootstrapped correlations. Substantial variability exists across model-specific correlations (Supplementary Figure 10b,c). Although the models generally agree on the signs of the correlations, the magnitudes of correlations differ strongly, possibly relating to different representations of land-atmosphere coupling and resulting differences in trends and initial ELI states (Supplementary Figure 5 and 9).

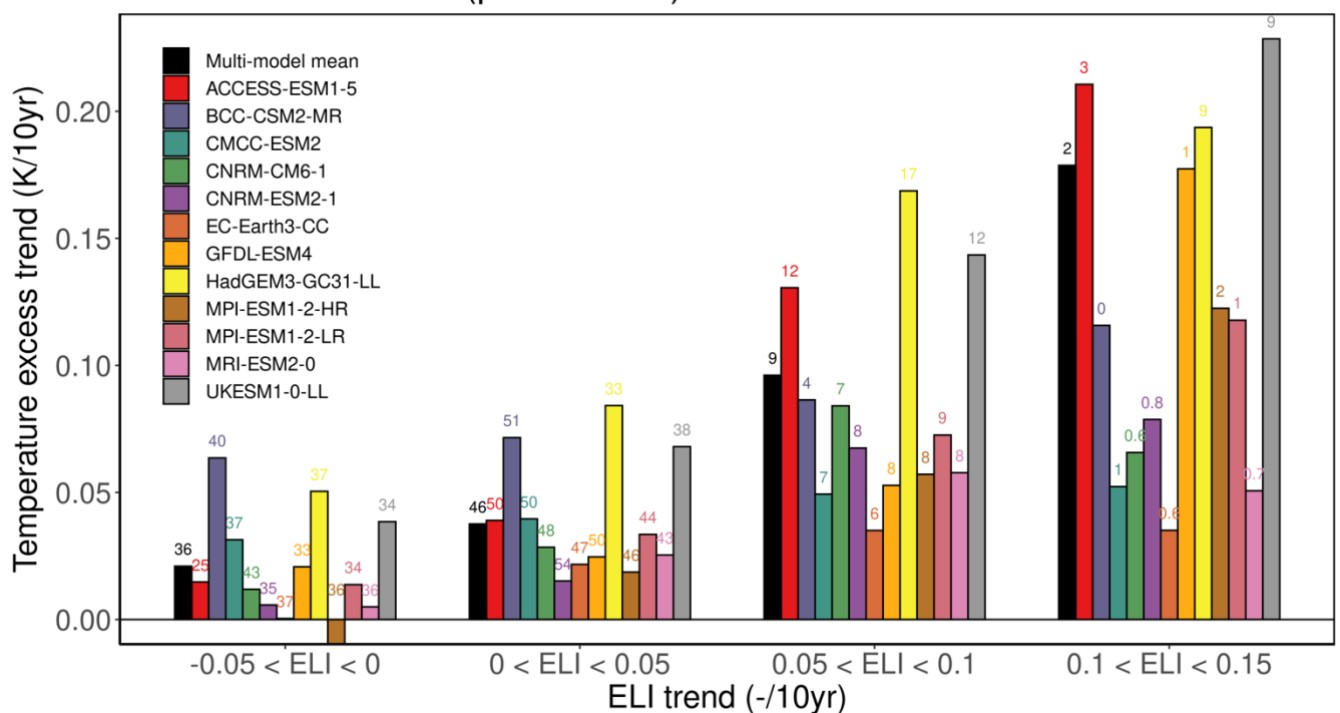

**Figure 6.** Temperature excess trends increase with stronger trends in ecosystem water limitation. The bars denote the multi-model mean and model-specific temperature excess trends (y-axis) binned according to their respective ELI trends (x-axis) for the multi-model mean trends (black) and all individual models (colors). The numbers display the fraction of warm vegetated land area in which respective temperature excess and ELI trends occur. These area fractions may not add up to 100%, because values outside of the defined bins on the x-axis are possible.

In order to further analyze the role of the magnitude of ELI trends for the coinciding temperature excess trends, we group the global grid cells with respect to their ELI trends and show the multi-model mean and model-specific temperature excess trends (Figure 6). Higher temperature excess trends correspond to stronger increasing ELI trends. Such strong increases in ELI indicate more often occurring water-limited conditions, potentially also during heat wave events, such that temperature excess gets more sensitive to ELI. Analyzing results from individual models shows that stronger ELI trends are associated with stronger trends in temperature excess in almost all models, albeit with substantial variability between individual models, owing to different representations and strength of land-atmosphere coupling.

## 4 Discussion

Our findings corroborate earlier research which demonstrated the relevance of soil moisture to (future) heat extremes via its control on surface flux partitioning based on idealized Earth system model experiments in which long-term soil moisture trends are artificially removed (Fischer et al., 2007; Lorenz et al., 2016; Schwingshackl et al., 2018; Seneviratne et al., 2006; Vogel et al., 2017, 2018). While our correlative analysis cannot establish the causal link nor disentangle the direction of causality between land surface dynamics and heat extremes to the same extent, it benefits from fully coupled simulations without artificial tweaking the water balances, such that it effectively complements the existing body of research. We note that temperature excess is not exclusively driven by land-atmosphere coupling, and the findings presented here merely stress the importance of considering ELI in this context.

While the correlation between ELI and heat wave temperatures is robust across models, we find substantial differences between individual models in terms of the strength of this link (e.g. Figure 2 and 6 and Supplementary Figures 6, 7 and 10). This could be related to a different representation of land-atmosphere interactions in general, which could be due to e.g. different soil moisture layers and depths, as well as different underlying soil and vegetation types. Additionally, models might use different vegetation water stress functions, some of which are poorly constrained by theory (De Kauwe et al., 2017; Martínez-de la Torre et al., 2019; Ukkola, Kauwe, et al., 2016). Further, not all models include dynamic vegetation, irrigation and land use change (Table 1). Another reason might be that measurements of soil moisture and terrestrial evaporation are scarce, such that large-scale observational constraints for these key quantities have been lacking and are only recently available following the advent of machine-learning techniques to efficiently interpolate global gridded datasets from the available in-situ measurements (Jung et al., 2019; O & Orth, 2021). Additionally, the vegetation's response to soil moisture drying is difficult

to capture due to heterogeneous soil and vegetation characteristics and limited observational constraints for rooting depths and soil moisture dynamics in respective soil layers. Next to those processes, the effects of ELI on temperature excess can be obscured by land use, circulation change and trends in incoming shortwave radiation (Supplementary Figure 4). Although disentangling such effects would be insightful, we consider a comprehensible analysis out of scope for this study. At the same time, the findings in this study are based on model-specific assumptions. Therefore, we advocate the need to reproduce the main findings in this study (Figure 1c, for example) with observation-based data to scrutinize the model-based findings in this study. However, despite apparent differences in processes represented in the models, we still find mostly significant positive correlations between temperature excess and ELI in most models (Supplementary Figure 6).

Further, despite the apparent difficulty that Earth System Models experience with representing soil moisture trends and related trends in land-atmosphere processes (Albergel et al., 2013; Berg et al., 2017; Berg & Sheffield, 2018; Greve et al., 2019), widespread shifts towards water limitation are robustly projected (Figure 1) (Denissen et al., 2022; Teuling, 2018; Ukkola et al., 2018). Further highlighting the complex nature of land-atmosphere interactions, we note that ecosystem water limitation is not only affected by climate, but also by changes in vegetation physiology (e.g. stomatal regulation) and structure (e.g. LAI) in response to increasing $CO_2$ ($CO_2$ fertilization) (Donohue et al., 2013; Ukkola, Prentice, et al., 2016; Walker et al., 2021; Zhu et al., 2016), which has also been shown to modulate heat extremes (Lemordant & Gentine, 2019). This way, changes of both $CO_2$ and climate jointly affect ELI which in turn influences heat wave magnitudes. Given this situation, future research should focus on the link between ELI and heat wave intensities using observation-based datasets, particularly as longer-term interpolations or reconstructions of key variables become available. This can help to corroborate model-based findings, and to constrain the variable relevance of ELI across models.

Finally, we focus on the intensity of the heat extremes by considering temperature only rather than more impact-relevant indices. Heat stress for humans is dependent not only on temperature, but also on wind speed and humidity (Buzan & Huber, 2020; Matthews, 2018). Through reduced evaporative cooling and increased entrainment of dry air from above the atmospheric boundary layer, the lethality of heat extremes above dry soils can be reduced (Wouters et al., 2022). In this study, we find an increasing temperature excess alongside increasing EF in 14% of the warm vegetated land area (Figure 2b), which suggests potentially higher heat stress than reflected by temperature alone as terrestrial evaporation can increase humidity and related lethality. On the other hand, combined hot and dry conditions can lead to increased wildfires (O et al., 2020) and can be associated with severe impacts on agriculture and infrastructure. In that perspective, our results on the correspondence between increased ecosystem water limitation and amplified heat waves confirm findings from Teuling (2018) indicating that droughts in Europe will become hotter under future warming. This is in line with future projections, suggesting that concurrent hot and dry extremes will continue to increase in future (Seneviratne et al., 2021; Vogel et al., 2020).

**5 Conclusion**

In conclusion, we show the ability of the land surface to modulate the intensity of future heat extremes. We focus on novel indices by focusing on ecosystem water limitation and the temperature excess between warm-season mean and maximum temperatures. In this context, the ELI is used to represent the nonlinear relationship between soil moisture and evaporative cooling, as it considers the effect of hydrometeorological anomalies on ecosystem response. This way, we find a widespread increase in temperature excess in ~75% of our study area. We identify several regions of interest where temperature excess is increasing more rapidly than the global mean. In large parts of these regions, these temperature excess increases jointly occur with trends towards ecosystem water limitation which lead to reduced evaporative cooling. Thereby, the relevance of trends in ecosystem water limitation for trends in temperature excess depends on (i) the magnitude of the ELI trends, which is largest in initially energy-limited and transitional areas, and (ii) the initial ELI regime as (maximum) temperatures are more sensitive to evaporative cooling in initially water-limited regions.

Finally, identifying regions where ELI trends and related evaporative cooling are important for future heat extremes can inform long-term adaptation strategies. Human activities play a key role here, as we can implement agricultural practices and/or tillage, irrigation and land cover management, afforestation and city greening to mitigate the impact of heat extremes (Schwaab et al., 2021; Sillmann et al., 2017).

**Data and code availability**

The CMIP6 model simulation data is freely available from the Earth System Grid Federation (ESGF) public data: https://aims2.llnl.gov/search/?project=CMIP6/. All the data used in this analysis is made publicly available in a data repository which can be accessed via Zenodo: https://zenodo.org/doi/10.5281/zenodo.11072826.

The scripts to acquire CMIP6 data are publicly available (https://github.com/TaufiqHassan/acccmip6) (Hassan, 2022). All the code written and used in this analysis are made available from a code repository on Zenodo: https://zenodo.org/doi/10.5281/zenodo.11073162.

**Author contributions**

R.O., A.J.T. and J.M.C.D. jointly designed the study. J.M.C.D. performed the analyses. All authors contributed to the writing of the paper, the discussion and interpretation of the results.

**Competing Interest Statement**

The authors declare no competing interests.

**Acknowledgements**

R.O. is supported through funding from the German Research Foundation (Emmy Noether Grant 391059971). We thank the respective climate modelling groups for making their model output available within the Coupled Model Intercomparison Project Phase 6 (CMIP6) ensemble. Further, I want to acknowledge the fruitful discussions within the Hydrosphere-Biosphere-Climate Interactions group in the Biogeochemical Integration Department of the Max Planck Institute for Biogeochemistry that have contributed to the interpretation of the results and design of the figures. A special thanks to Sujan Koirala for making

the scripts to download CMIP6 data from Google cloud CMIP6 public data publicly available and supporting whenever issues came up. Another special thanks to Ulrich Weber for downloading and aggregating the reanalysis data used in this study.

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
