# Peer review of "Intensified future heat extremes linked with increasing ecosystem water limitation"

_EGUsphere, 2023_

## Referee Comment (RC2)

**Review of "Intensified future heat extremes linked with increasing ecosystem water limitation" by Denissen *et al.*, 2023**

Using a small ensemble of CMIP6 simulations, the authors show that areas with increasing ecosystem water limitation tend to feature stronger warm season maximum temperature trends (compared to mean temperature changes). While the mechanisms behind this have long been known, most analyses focus on past changes and it is an interesting, well-designed study that I consider to be relevant for a broad audience. Nevertheless, I list a few suggestions below that could be helpful in further improving the manuscript.

**Comments**

1.) I am not convinced by the choice of "mrso" to indicate root-zone soil moisture. "mrso" is simply the total column soil moisture, and the actual depth that is represented varies from model to model and can easily exceed 2 meters (Qiao et al., 2022). In the Supplementary, it becomes clear that you use ERA5-Land soil moisture down to 100 cm (first 3 layers), and I think this is a good choice as the bottom layer extending to nearly 3m depth is arguably more uncertain. However, it would probably make sense to use the very same definition for the CMIP6 models, and not rely on the column soil moisture. 1m soil moisture could be calculated by using all layers within 100 cm and adding a fraction of the respective lowermost layer (e.g., 0.5 if it extends from 80 to 120 cm).

2.) I am quite surprised to see how few models seem to have all the required variables, especially since you only need them in monthly resolution. I get at least 40 different models (not simulations, as for some models such as, e.g., CanESM5, MPI-ESM-LR or MIROC6, there are dozens of initial condition ensemble members) for each variable, and while I did not check the overlap for all variables, I am absolutely sure that far more than 8 models remain. It should be close to or even more than 30…

I would also like to point out that according to Qiao et al. (2022), the BCC-CSM2-MR model constitutes a rather unfortunate "choice", as it does not perform well with regards to soil moisture. Moreover, to quote Qiao et al. (2022), "For deep soil moisture, the top-five best-performing models are CESM2, MPI-ESM1-2-LR, ACCESS-ESM1-5, CESM2-WACCM, and CNRM-ESM2-1, [...]", of which only CNRM-ESM2-1 is used here. While such evaluations are particularly challenging for variables that are hardly observed/measured and notoriously spatially inhomogeneous, I still think it is a pity that a) only few models were used in the first place, and b) that state of the art models such as CESM2 with plant hydraulics (see, e.g., Zhao et al., 2022) are not included. I thus

strongly encourage the authors to check an alternative data source if they cannot obtain the required variables for more than the 8 models used thus far.

3.) I appreciate that the authors state that land–atmosphere coupling does not necessarily account for all of the "temperature excess", but it also makes me wonder what else could contribute to stronger maximum than mean temperature trends. I agree that (changes in) advection could play a role, but I think there is another, perhaps even more important mechanism at play: in several regions around the world, aerosol emissions have decreased substantially and are projected to decrease further in the ongoing century. This results in more shortwave radiation reaching the surface compared to past decades due to higher atmospheric transmission, which noticeably alters the surface energy budget and hence near-surface temperatures (e.g., Nabat et al., 2014), particularly in the warm season when incoming shortwave radiation is typically highest. Maximum temperatures tend to occur between noon and late afternoon and are arguably closer related to incoming shortwave radiation than mean temperatures, which, during nighttime, are primarily governed by the longwave radiation budget (which is directly altered by anthropogenic greenhouse gas emissions and water vapor feedbacks). The study of Qian et al. (2011) supports this rationale by reporting that aerosol-related temperature effects mostly occur through (daytime) maximum temperatures. I would thus not be surprised if shortwave radiation changes — which can, of course, also be mediated by changes in cloudiness and not just aerosol absorption (although at least for central Europe, this aspect has been far less important since 1980; see, e.g., Wild et al., 2021) — also contributed to the temperature excess patterns shown in Fig. 1a. In some regions such as, e.g., China (Qian et al., 2011), India and central Africa, shortwave radiation has decreased in the last decades, so my example provided above should not be generalized. Showing downward shortwave radiation trends (rsds) for all models could be helpful to understand why areas where the sign of temperature excess and ELI trends is inconsistent.

**Additional comments**

- Some citations should be double-checked; e.g., "(Eyring et al., 2016))" comes with an additional right bracket.
- L. 85: I recommend changing "[...] please refer to Denissen et al. (Denissen et al., 2022)" to "please refer to Denissen et al. (2022)". Same thing for "from Teuling et al. (Teuling, 2018)" on L. 321.
- L. 167 onwards: "Moreover, ET is generally significantly correlated with both temperature excess and ELI, respectively, establishing the physical link between these quantities". The authors acknowledge themselves later on in the manuscript that their correlative analysis cannot establish causal links, so perhaps something like, e.g., "[...] , suggesting a physical link [...] " would be more appropriate.

- L. 200 onwards: ERA5-Land is an offline land surface model simulation that does *not* assimilate any observations. The meteorological forcing provided by ERA5 does indeed make use of data assimilation, but this is largely restricted to "classic" variables such as 2-meter temperature and humidity. Surface soil moisture data from scatterometers is also assimilated, but this only affects the top soil layer and does not help much with regards to root-zone soil moisture.
- L. 315: "[...] increased entrainment of dry air above the atmospheric boundary layer", I think rephrasing this to "[...] increased entrainment of dry air from above the [...]" or similar would be a good idea, the current version could be a bit confusing.

**References**

Pierre Nabat, P., Somot, S., Mallet, M., Sanchez-Lorenzo, A. & Wild, M (2014): Contribution of anthropogenic sulfate aerosols to the changing Euro-Mediterranean climate since 1980. *Geophys. Res. Lett.* **41**, 5605–5611. doi=10.1002/2014GL060798

To cite this article: Qian, Y., Leung, L. R., Ghan, S. J. & Giorgi, F. (2011): Regional climate effects of aerosols over China: modeling and observation. *Tellus B Chem. Phys. Meteorol.* **55**, 914–934, doi=10.3402/tellusb.v55i4.16379

Qiao, L., Zuo, Z. and Xiao, D. (2022): Evaluation of Soil Moisture in CMIP6 Simulations. *J. Clim.* **35**, 779–800, doi=10.1175/JCLI-D-20-0827.1

Wild, M., Wacker, S., Yang, S., Sanchez-Lorenzo, A (2021): Evidence for Clear-Sky Dimming and Brightening in Central Europe. *Geophys. Res. Lett.* **48**, e2020GL092216. doi=10.1029/2020GL092216

Zhao, M., A, G., Liu, Y. *et al.* (2022): Evapotranspiration frequently increases during droughts. *Nat. Clim. Chang.* 12, 1024–1030. doi=10.1038/s41558-022-01505-3

---

## Author Comment (AC1)

We are thankful to all reviewers for their valuable feedback which helped us to improve the manuscript. In response, aside from several minor corrections, we have introduced the following main changes to the paper:
- We have increased the amount of models considered in this study from 8 to 12.
- The ELI is now calculated with the soil moisture averaged over the top meter, which better represents effective water availability for terrestrial evaporation, as opposed to total column soil moisture.
- Hot spot region "NAS" has been moved northwards slightly and extended eastwards, as the regional pattern of largest changes in temperature excess has shifted slightly following the inclusion of additional models in the analysis.

As a result of these changes, the figures and main conclusions are even more pronounced or remain similar, which reflects the robustness of the methodology.
* * *
Using CMIP6 model projections, Denissen et al evaluate the co-occurrence of increasing trends in extreme temperature and increasing trends in ELI, a water-limitation metric. They find that these trends co-occur in many regions of the world especially in transitional and more energy limited regions. Therefore, more energy-limited locations are becoming more water-limited and experiencing more temperature extremes. This study is well done, carefully written, and concise which is always appreciated. I advocate for the use of ELI here which captures soil moisture and its nonlinear relation to energy fluxes. I find ELI to be a more direct variable to evaluate the questions here than soil moisture alone – something the authors could highlight more because it is a big strength compared to previous work.

My main criticism is the removal of many dryland regions, which I think are important for the message. I study the water, carbon, and energy cycles of these dry regions, including the influence of vegetation on the surface energy balance (for example, https://onlinelibrary.wiley.com/doi/abs/10.1111/gcb.16455; no expectation to cite). I am concerned that many of these regions are not fully included in the study and could bias overarching conclusions since they can respond so differently (see my #1 comment below). Nevertheless, I think it is a great study and ask the authors to consider several points.

-Andrew Feldman

 Main Comments

1) I find the condition in L114-115 to remove pixels at <0.5 m2/m2 of LAI is quite restrictive and removes many drylands, including the Sahel, most of China, and nearly all of Australia. These are key water limited regions to remove, especially in the context of heatwaves where these regions may be most vulnerable. Drylands have been deemed an important part of the climate system. Dryland vegetation also plays a critical role in the surface energy balance. See some studies here (with no expectation to cite) where meaningful dryland vegetation energy balance studies were conducted with different results from expectations:

https://www.science.org/doi/10.1126/science.abm9684

https://onlinelibrary.wiley.com/doi/abs/10.1111/gcb.16455

I suggest using a less restrictive condition. Or be very clear motivating why such a strict condition is used here to remove these dry places.

Thanks - we agree with this argumentation. In response we have made the LAI threshold less restrictive, filtering grid cells that have a monthly LAI lower than 0.2. Therefore, but also because we use more and a different set of CMIP6 models, we retain more dryland regions in the Sahel and in Australia. This is addressed in the following lines in the methodology

"Second, to additionally assure that we are investigating the active vegetation periods during the warm season, which would elicit vegetation responses to anomalies in energy and water supply affecting the surface flux partitioning, all months with $T_a < 10°C$ and Leaf Area Index (LAI) $< 0.2$ m2 m-2 are excluded from the analysis. Thereby, we disregard mainly grid cells in the most sparsely vegetated regions in Northern Africa and Western China and cold regions in the Northern latitudes, but retain major drylands including parts of the Sahel and the Australian interior (Supplementary Figure 2)."

[Figure]

**Supplementary Figure 2:** *Data points retained after masking. Columns denote the applied filtering procedures (from left to right: Ta < 10˚C, LAI < 0.5 and Ta < 10˚C & LAI < 0.5). Rows reflect the different individual models. The colors show the amount of values retained after filtering, where the maximum amount of values possible equals 3 hottest months per year over 120 years (360 data points). No data is available in the white regions.*

These water-limited dryland regions play an important role in temperature excess trends, as sensitivity of temperature excess trends to ELI in such regions is the highest (Figure 5d). We clarify this in the following lines in the abstract, the results and the conclusion.

"Sensitivity of temperature excess trends to ELI trends is highest in water-limited regions, such that in these regions relatively small ELI trends can amount to drastic temperature excess trends."

"Moving beyond trends we also analyze the sensitivity of decadal temperature excess with respect to ELI for energy-limited vs. transitional vs. water-limited areas and find the strongest relationship in the case of water-limited areas (Figure 5d), as evidenced by the largest increase

in temperature excess with ELI. This confirms that changes in water-limited areas temperature excess trends are most sensitive to ELI trends. This stresses that evaporative cooling in already arid drylands is even further reduced, increasingly limiting their ability to mitigate future heat extremes (Feldman et al., 2023)."

[Figure]

**Figure 5.** *Relation between temperature excess and ecosystem water limitation. a) Multi-model mean Ecosystem Limitation Index (1980 - 2010). Solid lines depict the time series of multi-model means inferred from globally (black) and regionally (colored) decadally averaged model simulations for b) temperature excess and c) Ecosystem Limitation Index. The classification is defined based on the model-specific mean ELI over 1980 - 2010 (Supplementary Figure 9): Energy limited (ELI < -0.2), transitional (-0.2 < ELI < 0.2) and water limited (ELI > 0.2). d) Points denote the global (black) and regional (colored) decadal multi-model means of ELI (x-axis) and temperature excess (y-axis), expressed as change since 1980. The lines denote linear regressions, with a shaded colored 95% confidence interval. Land grid cells that do not have complete time series for all models are excluded (white regions, Methods). Global and regional averages are weighted according to the surface area per grid cell.*

"Thereby, the relevance of trends in ecosystem water limitation for trends in temperature excess depends on (i) the magnitude of the ELI trends, which is largest in initially energy-limited and transitional areas, and (ii) the initial ELI regime as (maximum) temperatures are more sensitive to evaporative cooling in initially water-limited regions."

2) In support of this study, I think a huge advantage of this study is the use of ELI rather than soil moisture alone. This point is not clear in the study and I think it is one of the main points to make up front on why this complements existing literature so well. Most studies typically evaluate the question of how the land surface influences temperature extremes with soil moisture. However, because soil moisture is nonlinearily related to energy fluxes, it limits soil moisture's use to evaluate temperature by itself. A more important variable that captures this nonlinearity and soil moisture variability simultaneously is how water-limited versus energy limited a location is. ELI is one nice way to capture this (my variable of choice is time spent in the water-limited regime). I suggest making this over point clearer throughout.

We have further clarified the benefits of using ELI over soil moisture alone in the introduction, the results and the conclusion.

"In particular we use (i) a recently introduced ecosystem water stress index (Ecosystem Limitation Index (ELI), (Denissen et al., 2020)), a correlative index that evaluates directly the importance of water versus energy stress for terrestrial evaporation, thereby moving beyond the nonlinear relationship between soil moisture and evaporative cooling alone. Further, as this index directly captures evaporative cooling, it links more mechanistically with heat waves than general aridity or land-atmosphere coupling indices. Thereby other factors affecting water-limitation can be functionally addressed (e.g. groundwater, hydraulic failure as lag effect, $CO_2$). Further, the ELI can be used to pinpoint regime transitions, as positive values are indicative of water-limited conditions, while negative values denote ecosystem energy limitation."

"The sensitivity of temperature excess to ELI trends is expected to depend on the initial regime and can be explained through the nonlinear relationship between soil moisture and EF (Supplementary Figure 20 in Denissen et al., 2022; Seneviratne et al., 2010): In initially energy-limited grid cells (soil moisture exceeds critical soil moisture), ecosystems can sustain maximum EF, assuming sufficient available energy during the warm season. Hence, in such grid cells shifts towards water limitation, expressed by positive ELI trends or soil drying, do not amount to large changes in surface flux partitioning, nor in temperature excess, resulting in low sensitivity between ELI and temperature excess trends. In initially water-limited grid cells (soil moisture below critical soil moisture), further soil drying, or shifts towards water limitation, can reduce EF. This way, temperature excess trends are highly sensitive to ELI trends in water-limited grid cells. Transitional grid cells, which are characterized by a soil moisture regime that transitions periodically from below to above the critical moisture content, effectively switch between energy- and water-limited conditions frequently. As such, evaporative cooling and consequently temperature excess are periodically sensitive to increasing water limitation. In extremely dry and water-limited conditions, where soil moisture values approach the wilting point, hardly any moisture can be extracted from the soil, rendering vegetation activity and associated EF too low to provide ample evaporative cooling. As such, shifts towards

ecosystem water limitation should hardly decrease evaporative cooling further in extremely water-limited grid cells."

"In conclusion, we show the ability of the land surface to modulate the intensity of future heat extremes. We focus on novel indices by focusing on ecosystem water limitation and the temperature excess between warm-season mean and maximum temperatures. In this context, the ELI is used to represent the nonlinear relationship between soil moisture and evaporative cooling, as it considers the effect of hydrometeorological anomalies on ecosystem response."

3) Language and bias of thinking throughout seems to be about how ELI is influencing excess temperatures and that the direction of causality is from ELI to excess temperature. For example, see lines 275-276. Following that, it is nicely stated that this correlative analysis does not mean causality. However, I do suggest also noting in the discussion or elsewhere how excess temperature can influence ELI. This might help complete the loop on that discussion since I think the feedback in the opposite direction of heatwaves on ELI is also just as interesting and valuable. In other words, the authors might be limiting themselves in influencing the reader to think about ELI influencing on temperature extremes, when the other way around can give insights about sustaining heatwaves.

A more elaborate discussion on the direction of causality between ELI and temperature excess is added in the results section.

"This is evidenced by significant correlations in many areas (Figure 1c, Supplementary Figure 6), suggesting that increasing ELI contributes to hotter temperature extremes. As correlations cannot distinguish the direction of causality, we stress that hotter temperature extremes can in turn further dry out terrestrial vegetation, thereby increasing water limitation. Additionally, heat extremes and related hydraulic failure could lead to plant mortality (McDowell & Allen, 2015), limiting evaporative cooling even more. As such, these pathways further strengthen positive correlations between ELI and temperature excess."

[Figure]

**Figure 1.** *Similarity of global patterns of change in temperature excess and ecosystem water limitation. Multi-model means of trends based on decadal time series per respective CMIP6 model of a) temperature excess) and b) Ecosystem Limitation Index (ELI). c) Multi-model means of Kendall's rank correlation coefficient between model-specific time series of ELI and temperature excess. The insets display the fraction of the warm land area with positive or negative trends or correlations, respectively (at least 8 out of 12 models agreeing on the sign of the trend or correlation are hued darker). Stippling indicates that at least 8 out of 12 CMIP6 models agree on the sign of the trend or correlation. All trends and correlations are calculated over the warm season and are only displayed if at least 8 CMIP6 models have full time series available, such that white areas denote regions with no or insufficient data. The dashed boxes indicate regions of interest, which are regions where temperature excess increases are particularly rapid and spatially coherent: North and South America (NAM and SAM), Central Europe (CEU) and Northern Asia (NAS).*

[Figure]

**Supplementary Figure 6:** *Kendall's rank correlation coefficient between ecosystem water limitation and temperature excess per individual CMIP6 model (dots indicate significance: p < 0.05).*

4) Figure 3 is really neat. I think it could be a better facilitated display of results in Fig. 3 and lines 231-247 if the nonlinear ET-soil moisture (and maybe also ET-SWin) relationships are discussed/displayed more prominently. I think the authors are making claims about how EF is insensitive to water in energy limited regions and might become even insensitive at lower soil moisture in water-limited places. These would be better supported if the Budyko framework and/or EF-soil moisture relationships are introduced before these other points are made about Figure 3.

We now explain the sensitivity of temperature excess to ELI trends at the hand of the nonlinear relationship between SM and EF as described by Seneviratne et al (2010). Further, we refer to Supplementary material in Denissen et al. (2020), where we find a strong link between the fraction of days with soil moisture below critical soil moisture and ELI:

"The sensitivity of temperature excess to ELI trends is expected to depend on the initial regime and can be explained through the nonlinear relationship between soil moisture and EF (Supplementary Figure 20 in Denissen et al., 2022; Seneviratne et al., 2010): In initially energy-limited grid cells (soil moisture exceeds critical soil moisture), ecosystems can sustain maximum EF, assuming sufficient available energy during the warm season. Hence, in such

grid cells shifts towards water limitation, expressed by positive ELI trends or soil drying, do not amount to large changes in surface flux partitioning, nor in temperature excess, resulting in low sensitivity between ELI and temperature excess trends. In initially water-limited grid cells (soil moisture below critical soil moisture), further soil drying, or shifts towards water limitation, can reduce EF. This way, temperature excess trends are highly sensitive to ELI trends in water-limited grid cells. Transitional grid cells, which are characterized by a soil moisture regime that transitions periodically from below to above the critical moisture content, effectively switch between energy- and water-limited conditions frequently. As such, evaporative cooling and consequently temperature excess are periodically sensitive to increasing water limitation. In extremely dry and water-limited conditions, where soil moisture values approach the wilting point, hardly any moisture can be extracted from the soil, rendering vegetation activity and associated EF too low to provide ample evaporative cooling. As such, shifts towards ecosystem water limitation should hardly decrease evaporative cooling further in extremely water-limited grid cells."

5) This is a "devil's advocate" position, but something I worry about in studies using models to learn about land-atmosphere interactions is how much model biases in the relationships between soil moisture and energy fluxes (here EF) cause errors in results such as those presented here. I always look at CMIP or reanalysis based results and hope that ensemble means teach us emergent behavior of the land surface, rather than only give us back the potentially flawed relationship between soil moisture and EF that some models might have. This study is valuable in presenting the model results and also adds the dimension that projections can be made, which is not directly possible with observations. However, at least in the discussion, I suggest advocating for the main figures being reproduced in an observation-based study to test whether these model behaviors are reproduced in nature. For example, Figure 1c can be reproduced with satellite soil moisture and LST (or gridded air temperature) to give further support for the results here.

We agree with the reviewer. We assume that by taking a mean of many models with varying underlying assumptions on soil moisture and other stress functions. Even if this approach does not favor one model's flaws over the other, it is still based on a collection of model assumptions that need to be validated by observation-based studies. As time series of 120 years, as are used in this study, are not available from observation-based data sets, doing such an analysis would require a change in the methodology. Therefore, we think that this is out of scope for this analysis. However, we now advocate the need for observation-based analyses in the discussion, as the reviewer suggested.

"At the same time, the findings in this study are based on model-specific assumptions. Therefore, we advocate the need to reproduce the main findings in this study (Figure 1c, for example) with observation-based data to scrutinize the model-based findings in this study."

Further, we additionally advocate the use of observation-based data, as with time more and longer time series of observation-based variables will become available.

"This way, changes of both CO2 and climate jointly affect ELI which in turn influences heat wave magnitudes. Given this situation, future research should focus on the link between ELI and heat wave intensities using observation-based datasets, particularly as longer-term interpolations or reconstructions of key variables become available. This can help to

corroborate model-based findings, and to constrain the variable relevance of ELI across models."

6) There are many figures in the SI that are discussed extensively in the results. For example, Figure S6 about ET in lines 164-174 and Figure S8 in lines 198-211. I suggest moving them to the main text if they are pivotal parts of the manuscript.

We agree with the reviewer here and have moved supplementary figures 6 and 8 to the main text (now Figure 2 and 4).

[Figure]

*Figure 2. Global multi-model mean distribution and trends of Evaporative Fraction (EF). Multi-model mean of trends based on decadal time series per respective CMIP6 model of a) EF and b) Ecosystem Limitation Index (ELI). c) Multi-model mean of Kendall's rank correlation coefficient between model-specific time series of ELI and temperature excess. The insets display the fraction of the warm land area that with positive or negative trends or correlations, respectively (at least 8 out of 12 models agreeing on the sign of the trend or correlation are hued darker). Stippling indicates that at least 8 out of 12 CMIP6 models agree on the sign of*

*the trend or correlation. All trends and correlations are calculated over the three hottest months-of-year, defined as the 3 months–of-year which have the highest average temperature over 1980 - 2100. The dashed boxes indicate regions of interest.*

[Figure]

**Figure 4.** *Changes in global and regional temperature excess in concert with increasing ecosystem water limitation from CMIP6 models and ERA5-Land. Temporal evolution of a) temperature excess and of b) Ecosystem Limitation Index (ELI) globally and for the regions of interest. The black solid lines depict global and regional time series from the CMIP6 models, while the black dashed line represents ERA5-Land. The grey ribbon displays the envelope which encapsulates all the CMIP6 results. Global averages are calculated over land grid cells that have complete time series for all models and variables and are weighted according to the surface area per grid cell. The same mask is applied for CMIP6 models and ERA5-Land.*

Specific Comments

L12: note that the use of ecosystem (assuming both soil+vegetation) and vegetation are mentioned here which is making it unclear what the paper is about (is it only vegetation or soil+vegetation?). Potentially define what you mean by ecosystem here.

We clarify the use of ecosystem (both plant transpiration and soil evaporation) in the abstract.

"Heat extremes have severe implications for human health, ecosystems and the initiation of wildfires. Whereas they are mostly introduced by atmospheric circulation patterns, the intensity of heat extremes is modulated by terrestrial evaporation associated with soil moisture availability. Thereby, ecosystems provide evaporative cooling through plant transpiration and soil evaporation, which is reduced under drought stress."

L70: The "|" symbol indicates conditioning in mathematics/probability. It is unclear how it is being used in the correlation function "cor(Ta'|SWin',ET')." It sounds like the correlation is either between Ta and ET or Ta and SWin based on line 75. Therefore, I think the "|" symbol is being used to somehow indicate this potential alternation in the metric. However, one can also interpret that notation as the correlation of Ta' with ET' while conditioning (or binning) Ta' on SWin'. Can the authors be clearer about this notation? I know L85 says to refer to another study for details of ELI, but details like this should be shared here for completeness.

We explain the notation in the following lines:

"In this context, the | indicates the use of either Ta or SWin anomalies in the second term on the right hand side of Eq. 1, as ET in some regions is limited more strongly by lack of incoming shortwave radiation (Nemani et al., 2003) and in other regions more strongly by cold temperatures."

L95, Table 1: It might be worth noting what the difference in r1/r2 and f1/f2 mean since not all are the same in that column.

In the current selection there are only differences in f1/f2/f3, which are now explained in the caption of Table 1.

"*: in the CMIP6 members, or variants, differences exist in the forcing index (f). This index number indicates the forcing used for the respective realization and can be used to distinguish between CMIP6-recommended or other forcing data sets. Which forcing dataset f represents is defined per model."

L114-115: The LAI condition at 0.5 m2/m2 might be overly restrictive and remove many drylands from the analysis that are important facets of the global climate.

See answer to 1).

L115: Central Africa? Do you mean East Africa?

We have adapted to "Northern Africa".

L118: It should be the sum of radiative components minus the ground heat flux (G) (or Rn-G).

We decided to neglect ground heat flux in our analysis, as we do not expect that it can significantly influence trends in ecosystem water limitation or excess heat. It is more relevant on a diurnal scale of course.

L150-152: This statement is tough to follow. This is only referring to the second term on the right side of Equation 1 or the energy limited component of ELI? I was thinking that water-limitation should be a big component in the tropics (but it looks like water-limitation is not considered in Fig. S1)

We have adapted the writing to more clearly explain that this indeed concerns only the second term on the right hand side of Equation 1:

"cor(SM',ET') is a proxy for water limitation, whereas cor(Ta' | SWin',ET') is a proxy for energy limitation. In this context, the | indicates the use of either Ta or SWin anomalies in the second term on the right hand side of Eq. 1, as ET in some regions is limited more strongly by lack of incoming shortwave radiation (Nemani et al., 2003) and in other regions more strongly by cold temperatures. Therefore, we test for each grid cell which energy proxy yields the highest correlation with ET (cor(Ta',ET') vs. cor(SWin',ET')), and is hence most relevant in this location, to then use it in the computation of ELI in the respective grid cell (Supplementary Figure 1)."

[Figure]

**Supplementary Figure 1:** *Spatial distribution of the sum of models that are temperature-controlled. Colors show the sum of models for which cor(Ta',ET') > cor(SWin',ET') over 1980 – 2100.*

L157-158: It could be the other way around where temperature extremes contribute to increasing ELI.

See answer to 3).

L158-L160: With removal of many drylands and some opposing results in these locations (see my comment 1), it would be worth discussing further what physical processes cause these regions to differ.

We have added Supplementary Figure 4, which shows that in regions with insignificant or negative correlations between ELI and temperature excess, trends in incoming shortwave radiation are generally also negative. We discuss this in lines XXX-XXX

Lines XXX-XXX
"Further deviations from a positive relationship between temperature excess and ELI might result from alternative processes such as (changes in) advection of warm air masses through large-scale circulation patterns and changes in incoming shortwave radiation (Supplementary Figure 4)."

[Figure]

*Supplementary Figure 4:* *Multi-model mean trend in incoming shortwave radiation based on decadal time series per respective CMIP6 model. The insets display the fraction of the warm land area with positive or negative, respectively (at least 8 out of 12 models agreeing on the sign of the trend are hued darker). Stippling indicates that at least 8 out of 12 CMIP6 models agree on the sign of the. All trends are calculated over the warm season and are only displayed if at least 8 CMIP6 models have full time series available, such that white areas denote regions with no or insufficient data. The dashed boxes indicate regions of interest, which are regions where temperature excess increases are particularly rapid and spatially coherent: North and South America (NAM and SAM), Central Europe (CEU) and Northern Asia (NAS) (see Figure 1).*

---

## Author Comment (AC2)

We are thankful to all reviewers for their valuable feedback which helped us to improve the manuscript. In response, aside from several minor corrections, we have introduced the following main changes to the paper:
- We have increased the amount of models considered in this study from 8 to 12.
- The ELI is now calculated with the soil moisture averaged over the top meter, which better represents effective water availability for terrestrial evaporation, as opposed to total column soil moisture.
- Hot spot region "NAS" has been moved northwards slightly and extended eastwards, as the regional pattern of largest changes in temperature excess has shifted slightly following the inclusion of additional models in the analysis.

As a result of these changes, the figures and main conclusions are even more pronounced or remain similar, which reflects the robustness of the methodology.
* * *
Using a small ensemble of CMIP6 simulations, the authors show that areas with increasing ecosystem water limitation tend to feature stronger warm season maximum temperature trends (compared to mean temperature changes). While the mechanisms behind this have long been known, most analyses focus on past changes and it is an interesting, well-designed study that I consider to be relevant for a broad audience. Nevertheless, I list a few suggestions below that could be helpful in further improving the manuscript.

Main comments
1.) I am not convinced by the choice of "mrso" to indicate root-zone soil moisture. "mrso" is simply the total column soil moisture, and the actual depth that is represented varies from model to model and can easily exceed 2 meters (Qiao et al., 2022). In the Supplementary, it becomes clear that you use ERA5-Land soil moisture down to 100 cm (first 3 layers), and I think this is a good choice as the bottom layer extending to nearly 3m depth is arguably more uncertain. However, it would probably make sense to use the very same definition for the CMIP6 models, and not rely on the column soil moisture. 1m soil moisture could be calculated by using all layers within 100 cm and adding a fraction of the respective lowermost layer (e.g., 0.5 if it extends from 80 to 120 cm).

We agree with the reviewer here. We have recomputed the ELI and remade all figures with soil moisture from layers averaged over the top meter of the soil. Using this root-zone soil moisture is more representative for the water availability that ecosystems experience. As a result of these changes, the figures and main conclusions are even more pronounced or remain similar, which reflects the robustness of the methodology.

[revised manuscript text omitted]

2.) I am quite surprised to see how few models seem to have all the required variables, especially since you only need them in monthly resolution. I get at least 40 different models (not simulations, as for some models such as, e.g., CanESM5, MPI-ESM-LR or MIROC6, there are dozens of initial condition ensemble members) for each variable, and while I did not check the overlap for all variables, I am absolutely sure that far more than 8 models remain. It should be close to or even more than 30...

I would also like to point out that according to Qiao et al. (2022), the BCC-CSM2-MR model constitutes a rather unfortunate "choice", as it does not perform well with regards to soil moisture. Moreover, to quote Qiao et al. (2022), "For deep soil moisture, the top-five best-performing models are CESM2, MPI-ESM1-2-LR, ACCESS-ESM1-5, CESM2-WACCM, and CNRM-ESM2-1, [...]", of which only CNRM-ESM2-1 is used here. While such evaluations are particularly challenging for variables that are hardly observed/measured and notoriously spatially inhomogeneous, I still think it is a pity that a) only few models were used in the first place, and b) that state of the art models such as CESM2 with plant hydraulics (see, e.g., Zhao et al., 2022) are not included. I thus
strongly encourage the authors to check an alternative data source if they cannot obtain the required variables for more than the 8 models used thus far.

Retrieving data from the Earth System Grid Federation (https://aims2.llnl.gov/search/?project=CMIP6/) instead of the Google cloud CMIP6 public data has led to a larger sample of 12 CMIP6 models that could be retrieved. These are the only models that meet the criteria described in the methodology (see lines below). The biggest

bottlenecks that prevented obtaining an even larger number of CMIP6 models were the unavailability of total water content per soil layer (mrsol), which excluded CIESM, HadGEM3-GC31-MM, INM-CM4-8, INM-CM5-0 and MIROC-ES2H, and/or unavailability of maximum daily temperature (tasmax), which excluded CESM2, CESM2-WACCM, CMCC-CM2-SR5, EC-Earth3-Veg and EC-Earth3-Veg-LR. Further, amongst the selected models, we have increased the amount of models with a better representation of deep soil moisture.

"We only selected models that provide i) historical (1980 - 2015) and "worst-case" SSP5-8.5 (2015 - 2100 (O'Neill et al., 2016)) simulations, ii) the necessary variables (Table 1) and iii) sufficient spatial (2°x2° or finer grid cell resolution) and temporal (monthly) resolutions."

3.) I appreciate that the authors state that land–atmosphere coupling does not necessarily account for all of the "temperature excess", but it also makes me wonder what else could contribute to stronger maximum than mean temperature trends. I agree that (changes in) advection could play a role, but I think there is another, perhaps even more important mechanism at play: in several regions around the world, aerosol emissions have decreased substantially and are projected to decrease further in the ongoing century. This results in more shortwave radiation reaching the surface compared to past decades due to higher atmospheric transmission, which noticeably alters the surface energy budget and hence near-surface temperatures (e.g., Nabat et al., 2014), particularly in the warm season when incoming shortwave radiation is typically highest. Maximum temperatures tend to occur between noon and late afternoon and are arguably closer related to incoming shortwave radiation than mean temperatures, which, during nighttime, are primarily governed by the longwave radiation budget (which is directly altered by anthropogenic greenhouse gas emissions and water vapor feedbacks). The study of Qian et al. (2011) supports this rationale by reporting that aerosol-related temperature effects mostly occur through (daytime) maximum temperatures. I would thus not be surprised if shortwave radiation changes — which can, of course, also be mediated by changes in cloudiness and not just aerosol absorption (although at least for central Europe, this aspect has been far less important since 1980; see, e.g., Wild et al., 2021) — also contributed to the temperature excess patterns shown in Fig. 1a. In some regions such as, e.g., China (Qian et al., 2011), India and central Africa, shortwave radiation has decreased in the last decades, so my example provided above should not be generalized. Showing downward shortwave radiation trends (rsds) for all models could be helpful to understand why areas where the sign of temperature excess and ELI trends is inconsistent.

We agree with the reviewer. We have inserted the multi-model mean incoming shortwave radiation trends in Figure 1b and show model-specific incoming shortwave radiation trends in Supplementary Figure 4. We have elaborate on incoming shortwave radiation trends in the following lines in the results section.

"There is a widespread increase in incoming shortwave radiation in about 71% of the warm vegetated land area, with high inter-model models agreement (Supplementary Figure 4), which can directly affect near-surface temperature through the surface energy balance. These trends could result from projected decreases in aerosol emissions (Nabat et al., 2014), or from changes in cloud cover. As daily maxima of incoming shortwave radiation roughly co-occur with daily temperature maxima, increased incoming shortwave radiation links more strongly to increased in maximum temperatures rather than mean temperatures (Qian et al., 2011), which are more strongly governed by the longwave radiation budget."

"Further deviations from a positive relationship between temperature excess and ELI might result from alternative processes such as (changes in) advection of warm air masses through large-scale circulation patterns and changes in incoming shortwave radiation (Supplementary Figure 4)."

[Figure]

*Supplementary Figure 4: Multi-model mean trend in incoming shortwave radiation based on decadal time series per respective CMIP6 model. The insets display the fraction of the warm land area with positive or negative, respectively (at least 8 out of 12 models agreeing on the sign of the trend are hued darker). Stippling indicates that at least 8 out of 12 CMIP6 models agree on the sign of the. All trends are calculated over the warm season and are only displayed if at least 8 CMIP6 models have full time series available, such that white areas denote regions with no or insufficient data. The dashed boxes indicate regions of interest, which are regions where temperature excess increases are particularly rapid and spatially coherent: North and South America (NAM and SAM), Central Europe (CEU) and Northern Asia (NAS) (see Figure 1).*

Additional comments
- Some citations should be double-checked; e.g., "(Eyring et al., 2016))" comes with an additional right bracket.

All double brackets were checked and removed if possible.

- L. 85: I recommend changing "[...] please refer to Denissen et al. (Denissen et al., 2022)" to "please refer to Denissen et al. (2022)". Same thing for "from Teuling et al. (Teuling, 2018)" on L. 321.

We have done as the reviewer suggested.

- L. 167 onwards: "Moreover, ET is generally significantly correlated with both temperature excess and ELI, respectively, establishing the physical link between these quantities". The authors acknowledge themselves later on in the manuscript that their correlative analysis cannot establish causal links, so perhaps something like, e.g., "[...] , suggesting a physical link [...] " would be more appropriate.

We have done as the reviewer suggested.

- L. 200 onwards: ERA5-Land is an offline land surface model simulation that does not assimilate any observations. The meteorological forcing provided by ERA5 does indeed make use of data assimilation, but this is largely restricted to "classic" variables such as 2-meter temperature and humidity. Surface soil moisture data from scatterometers is also assimilated, but this only affects the top soil layer and does not help much with regards to root-zone soil moisture.

We have adjusted the discussion accordingly:

"Note that ERA5-Land is only indirectly supported by data assimilation, as meteorological forcing from ERA5 assimilates observations only for 2m temperature, relative humidity and surface soil moisture. Therefore, temperature excess benefits more directly from data assimilation than ELI, which is based on ET and (root-zone) soil moisture which are not readily observed across the globe."

- L. 315: "[...] increased entrainment of dry air above the atmospheric boundary layer", I think rephrasing this to "[...] increased entrainment of dry air from above the [...]" or similar would be a good idea, the current version could be a bit confusing.

We have done as the reviewer suggested.

---

## Referee Report (RR1)

**Review of Denissen et al., round II**

I thank the authors for addressing each of my questions and suggestions in detail and agree with their responses, and therefore only reply to selected points below.

3.) [....]

We agree with the reviewer. We have inserted the multi-model mean incoming shortwave **radiation trends in Figure 1b** and show model-specific incoming shortwave radiation trends in Supplementary Figure 4.

We have elaborated on incoming shortwave radiation trends in the following lines in the results section (lines 181 – 186 and 204 - 206).
- Lines 181 – 186 "There is a widespread increase in incoming shortwave radiation in about 71% of the warm vegetated land area, with high inter-model agreement (Supplementary Figure 4), which can directly affect near-surface temperature through the surface energy balance. These trends could result from projected decreases in aerosol emissions (Nabat et al., 2014), or from changes in cloud cover. As daily maxima of incoming shortwave radiation roughly co-occur with daily temperature maxima, increased incoming shortwave radiation links more strongly to increased in maximum temperatures rather than mean temperatures (Qian et al., 2011), which are more strongly governed by the longwave radiation budget."
- Lines 204 - 206 "Further deviations from a positive relationship between temperature excess and ELI might result from alternative processes such as (changes in) advection of warm air masses through large-scale circulation patterns and changes in incoming shortwave radiation (Supplementary Figure 4)."

I suspect the response to my comment is no longer consistent with the revised manuscript, as I cannot identify any downward shortwave radiation trends in Fig. 1b. I have two more remarks here, (i) "These trends could result from projected decreases in aerosol emissions (Nabat et al., 2014), or from changes in cloud cover.", these effects *could* be separated by comparing all-sky (rsds) to clear-sky (rsdscs) downward shortwave radiation trends, but I don't think this is necessary for the presented analysis. (ii), currently, the reader is informed that the link between ELI and temperature excess could be partly masked by "alternative processes such as [...] changes in incoming shortwave radiation". Since this is a purely correlational analysis, however, I think it is also possible that we overestimate the role of ELI in causing temperature excess for the same reasons, since shortwave radiation definitely matters quite a bit as a driver of maximum temperature (e.g., Schwingshackl et al., 2018). I would like to leave it up to the authors whether they mention this caveat, and thank them for incorporating a downward shortwave radiation analysis.

- Some citations should be double-checked; e.g., "(Eyring et al., 2016))" comes with an additional right bracket.

All double brackets were checked and removed if possible.

Really not trying to be pedantic here, but I came across some new ones while reading the revised manuscript:
L. 69: ",(Denissen et al., 2020)),"
L. 308: ", (Denissen et al., 2022; Seneviratne et al., 2010)):"

**Additional comments:**
- Fig. 4: I am a bit puzzled by the fact that, although the observation-derived estimate (ERA5-Land) shows rather stark ELI changes compared to the CMIP6 model subset, this doesn't manifest with regards to temperature excess. As such, I am not sure whether this newly introduced 'temperature excess' is something that should already clearly emerge from historical, observation-based data. Of course, internal climate variability likely plays an important role here, but it seems quite difficult to reconcile these results given that for 4 out of 5 regions, ERA5-Land leaves the model envelope with regards to ELI, but stays well within concerning temperature excess except for the one region (NAM) where there is only a weak ELI signal. Based on this, I would argue that the picture is quite clear with regards to model projections until the end of the ongoing century,, but far less so in terms of historical data.
- Fig. 4a, legend: it still says ERA5
- Fig. 4b: units (K) seem odd

---

## Author Response (AR2)

We thank the reviewer for further comments, which improve the quality of the manuscript.

Review of Denissen et al., round II

I thank the authors for addressing each of my questions and suggestions in detail and agree with their responses, and therefore only reply to selected points below.

3.) [....]

We agree with the reviewer. We have inserted the multi-model mean incoming shortwave radiation trends in Figure 1b and show model-specific incoming shortwave radiation trends in Supplementary Figure 4.
We have elaborated on incoming shortwave radiation trends in the following lines in the results section (lines 181 – 186 and 204 - 206).
- Lines 181 – 186 "There is a widespread increase in incoming shortwave radiation in about 71% of the warm vegetated land area, with high inter-model agreement (Supplementary Figure 4), which can directly affect near-surface temperature through the surface energy balance. These trends could result from projected decreases in aerosol emissions (Nabat et al., 2014), or from changes in cloud cover. As daily maxima of incoming shortwave radiation roughly co-occur with daily temperature maxima, increased incoming shortwave radiation links more strongly to increased in maximum temperatures rather than mean temperatures (Qian et al., 2011), which are more strongly governed by the longwave radiation budget."
- Lines 204 - 206 "Further deviations from a positive relationship between temperature excess and ELI might result from alternative processes such as (changes in) advection of warm air masses through large-scale circulation patterns and changes in incoming shortwave radiation (Supplementary Figure 4)."

I suspect the response to my comment is no longer consistent with the revised manuscript, as I cannot identify any downward shortwave radiation trends in Fig. 1b. I have two more remarks here, (i) "These trends could result from projected decreases in aerosol emissions (Nabat et al., 2014), or from changes in cloud cover.", these effects could be separated by comparing all-sky (rsds) to clear-sky (rsdscs) downward shortwave radiation trends, but I don't think this is necessary for the presented analysis. (ii), currently, the reader is informed that the link between ELI and temperature excess could be partly masked by "alternative processes such as [...] changes in incoming shortwave radiation". Since this is a purely correlational analysis, however, I think it is also possible that we overestimate the role of ELI in causing temperature excess for the same reasons, since shortwave radiation definitely matters quite a bit as a driver of maximum temperature (e.g., Schwingshackl et al., 2018). I would like to leave it up to the authors whether they mention this caveat, and thank them for incorporating a downward shortwave radiation analysis.

(i) We agree with the reviewer here, and although further disentangling these drivers is interesting, it is out of scope for this paper.
(ii) We adapted the following sentence on lines 204-207:

"Further deviations from a positive relationship between temperature excess and ELI might result from alternative processes such as (changes in) advection of warm air masses through

large-scale circulation patterns, while positive relationships could be exaggerated by changes in incoming shortwave radiation (Supplementary Figure 4)."

 - Some citations should be double-checked; e.g., "(Eyring et al., 2016))" comes with an additional right bracket.

All double brackets were checked and removed if possible.

Really not trying to be pedantic here, but I came across some new ones while reading the revised manuscript:

L. 69: ",(Denissen et al., 2020)),"
L. 308: ", (Denissen et al., 2022; Seneviratne et al., 2010)):"

We apologize for overlooking the remaining double brackets. We have replaced them as follows:

L. 69 "...a recently introduced ecosystem water stress index: the Ecosystem Limitation Index, or ELI (Denissen et al.,2020). This is a correlative …"
L. 308 "... the nonlinear relationship between soil moisture and EF (Denissen et al., 2022; Seneviratne et al., 2010).

Additional comments:
- Fig. 4: I am a bit puzzled by the fact that, although the observation-derived estimate (ERA5-Land) shows rather stark ELI changes compared to the CMIP6 model subset, this doesn't manifest with regards to temperature excess. As such, I am not sure whether this newly introduced 'temperature excess' is something that should already clearly emerge from historical, observation-based data. Of course, internal climate variability likely plays an important role here, but it seems quite difficult to reconcile these results given that for 4 out of 5 regions, ERA5-Land leaves the model envelope with regards to ELI, but stays well within concerning temperature excess except for the one region (NAM) where there is only a weak ELI signal. Based on this, I would argue that the picture is quite clear with regards to model projections until the end of the ongoing century,, but far less so in terms of historical data.

We agree with the reviewer here and address the difference in coupling between ELI and temperature excess in lines 267-273:

"In this historical time period and across most regions of interest, the CMIP6 trends for both temperature excess and ELI are generally more positive than negative, which corroborates a positive relationship between the two, as is also seen further into the future (Figure 3). This relationship is weaker in the observation-based estimate from ERA5-Land, where temperature excess mostly stays within the multi-model envelope and only increases monotonically in SAM, while ELI exceeds the multi-model envelope and increases in all regions of interest except NAM. This indicates a different coupling between ELI and temperature excess in ERA5-Land than in the CMIP6 models, which should be further investigated in the future."

- Fig. 4a, legend: it still says ERA5

Adapted to "ERA5-Land"

- Fig. 4b: units (K) seem odd

Adapted to "(-)"

Adapted to "ERA5-Land"

- Fig. 4b: units (K) seem odd

Adapted to "(-)"